# Regionalized tissue fluidization is required for epithelial gap closure during insect gastrulation

Akanksha Jain [1,2], Vladimir Ulman[1,3], Arghyadip Mukherjee [4], Mangal Prakash[1,5], Marina B. Cuenca [1], Lokesh G. Pimpale [1,6], Stefan Münster[1,4,5,6], Robert Haase[1,5], Kristen A. Panfilio [7,8], Florian Jug [1,5], Stephan W. Grill[1,5,6,9], Pavel Tomancak [1,3 ✉] & Anastasios Pavlopoulos [10,11 ✉]

Many animal embryos pull and close an epithelial sheet around the ellipsoidal egg surface during a gastrulation process known as epiboly. The ovoidal geometry dictates that the epithelial sheet first expands and subsequently compacts. Moreover, the spreading epithelium is mechanically stressed and this stress needs to be released. Here we show that during extraembryonic tissue (serosa) epiboly in the insect *Tribolium castaneum*, the non-proliferative serosa becomes regionalized into a solid-like dorsal region with larger non-rearranging cells, and a more fluid-like ventral region surrounding the leading edge with smaller cells undergoing intercalations. Our results suggest that a heterogeneous actomyosin cable contributes to the fluidization of the leading edge by driving sequential eviction and intercalation of individual cells away from the serosa margin. Since this developmental solution utilized during epiboly resembles the mechanism of wound healing, we propose actomyosin cable-driven local tissue fluidization as a conserved morphogenetic module for closure of epithelial gaps.

[1] Max-Planck-Institute of Molecular Cell Biology and Genetics, Dresden, Germany. [2] Technische Universität Dresden, Dresden, Germany. [3] IT4Innovations, Technical University of Ostrava, Ostrava, Czech Republic. [4] Max-Planck-Institute for the Physics of Complex Systems, Dresden, Germany. [5] Center for Systems Biology, Dresden, Germany. [6] Biotechnology Center, TU Dresden, Dresden, Germany. [7] Institute for Zoology: Developmental Biology, University of Cologne, Cologne, Germany. [8] School of Life Sciences, University of Warwick, Coventry, UK. [9] Cluster of Excellence Physics of Life, TU Dresden, Dresden, Germany. [10] Janelia Research Campus, Howard Hughes Medical Institute, Ashburn, VA, USA. [11] Institute of Molecular Biology and Biotechnology, Foundation for Research and Technology-Hellas, Heraklion, Greece. ✉email: tomancak@mpi-cbg.de; a.pavlopoulos@imbb.forth.gr

Epiboly is one of the hallmark morphogenetic movements during animal gastrulation[1]. It involves spreading of an epithelial sheet over the spherical or ellipsoidal egg. The sheet eventually forms a continuous layer that entirely surrounds the embryo and the yolk sac. During this morphogenetic event,

fundamental geometrical and mechanical problems arise. First, in order to cover the entire egg, the epithelium has to expand in surface area. However, once the egg equator is reached, the expanding tissue must also undergo a regional compaction at its leading edge in order to seal seamlessly at the bottom of the sphere

**Fig. 1 Quantitative analysis of *Tribolium* serosa expansion. a** Schematic depiction of the geometric constraints experienced by a tissue expanding over a spherical yolk cell. The leading edge undergoes an area increase followed by an area decrease after it crosses the equator. **b** Illustrations of the stages of *Tribolium* embryogenesis from cellular blastoderm to serosa window closure. **c** 3D renderings of a *Tribolium* embryo expressing the fluorescent H2A-eGFP nuclear marker reconstructed from a multi-view time-lapse SPIM recording. The embryo is shown from the lateral and ventral views at the six reference stages corresponding to the schematics in **b**. All imaged embryos in this and other panels are shown with anterior to the left, and all time stamps are in hh: mm. Scale bar is 50 μm. $N = 1$ (2 datasets available). **d** 2D cartographic projections at reference stages of a 4D SPIM recording of a *Tribolium* embryo expressing EFA-nGFP. The extent of the serosal tissue is highlighted in turquoise. Scale bar is approximately 100 μm (see "Methods"). $N = 1$. **e** The area of the serosal tissue calculated from cartographic projections of 4D SPIM recordings. The data are normalized to the total serosa area at Stage 5 in each case. For every stage, the total serosa area is calculated for all time points between two consecutive stages in three different embryos and plotted as a distribution. Plots in this and all other panels indicate the median with a thick line, the mean with a black dot, and the standard deviation (s.d.) with the thin error bars. $N = 3$. **f** Comparison of the distributions of apical areas of cells sampled from confocal recordings of *Tribolium* embryos expressing the cortical LifeAct-eGFP actin marker at reference stages labeled according to **b**. The number of cells ($n$) and the number of embryos ($N$) sampled at different stages were in the dorsal region Stage 0 $n = 58$ and $N = 6$, Stage 1 $n = 116$ and $N = 11$, Stage 2 $n = 66$ and $N = 9$, Stage 3 $n = 39$ and $N = 6$, Stage 4 $n = 76$ and $N = 10$, and in the ventral region Stage 3 $n = 52$ and $N = 7$. The normal distribution of the data was tested using Shapiro–Wilk test. Distributions were compared using the non-parametric two-sided Wilcoxon Rank-Sum test (same in all figures unless stated otherwise). $p$ Values between 0.05 and 0.01 are labeled with single asterisk (*), 0.009–0.001 are labeled with double asterisks (**), <0.001 with triple asterisks (***), and ns signifies a non-significant $p$ value (same in all figures). **g** Cartographic projections at reference stages of a transgenic embryo labeled with LifeAct-eGFP and reconstructed from a multi-view SPIM recording. All serosal cell in each projection were segmented automatically, curated manually, and color coded according to their apical cell area. Red boxes indicate the approximate regions from which cells sampled in confocal datasets were quantified in **f**. $N = 1$.

(Fig. 1a). Studies in fish showed that the tissue spreading is mediated by changes in cell shape, cell number, and cell arrangement coupled to constriction of an actomyosin ring in the yolk at the leading edge of the sheet[2–5]. However, it remains unclear whether pulling forces at the leading edge expand cells uniformly throughout the tissue and how cells behave at the leading edge that needs to compact. Second, spreading over a sphere induces mechanical stress in the tissue. In zebrafish, mechanical stress during epibolic expansion is released by oriented cell divisions in the tissue[4]. In other epiboly systems however, cell division does not occur, and thus other unknown mechanisms have to alleviate built-up stress.

For example, in many insect taxa, the developing embryo is completely surrounded by a protective epithelial cell layer of extraembryonic fate called the serosa[6–10]. In the red flour beetle, *Tribolium castaneum*, extraembryonic serosal cells are initially specified as an anterior cap of the cellular blastoderm, which subsequently spreads over the gastrulating embryonic part of the blastoderm[11]. This process occurs in the complete absence of serosal cell division (Fig. 1b). The spreading serosal tissue expands over the posterior pole and eventually closes ventrally over the contracting embryo in a process known as serosa window closure[12–14]. However, it is not understood how the leading serosal cells at the rim of the serosa window achieve final compaction. It is also unknown if and how mechanical tension arises and gets released in the serosal tissue during spreading.

To address these questions, we use the *Tribolium* serosa epiboly and closure as a model to understand how the mechanical properties of serosal cells promote wrapping of a non-dividing epithelial sheet around an ellipsoidal egg. We find that serosal tissue becomes mechanically regionalized along the dorsal–ventral axis and that its ventral closure is facilitated by a local, actomyosin-cable-mediated fluidization at the leading edge.

## Results

### *Tribolium* serosa undergoes inhomogeneous expansion during epiboly. To visualize serosa epiboly, we imaged transgenic embryos expressing a nuclei-marking enhanced green fluorescent protein (eGFP) with multi-view light-sheet microscopy (Fig. 1c and Supplementary Movie 1). Taking advantage of the serosa's topology as a superficial egg layer, we unwrapped the three-dimensional (3D) data into two-dimensional (2D) cartographic time-lapse projections and segmented the serosal part of the blastoderm tissue[15] (Fig. 1d, Supplementary Fig. 1A–D, and

Supplementary Movies 2 and 11). The serosa covered initially about 35% of the egg surface and spread to cover 100% of the surface (Fig. 1e). In order to examine the expansion at the cellular level, we imaged embryos expressing LifeAct-eGFP that labels cortical F-actin[13,16] and segmented the apical surface of all serosal cells at the five reference stages (Fig. 1b) during serosa expansion (Fig. 1f, g). The results showed that the ~3-fold expansion in serosal tissue surface area was mirrored by a ~3-fold expansion of the apical area of serosal cells from Stage 1 to Stage 4 (Fig. 1f). Strikingly, serosal cells did not expand uniformly: at Stage 3, the apical area of ventral cells in the vicinity of the serosa window was on average 29% smaller compared to dorsal cells (Fig. 1f, g and Supplementary Movie 11). We conclude that serosa epiboly exhibits inhomogeneous apical cell area expansion in order to accommodate the ventral area compaction required by the elliptical geometry of the egg.

**Ventral leading edge of the serosa exhibits local tissue fluidization.** An alternative but not mutually exclusive mechanism to achieve ventral area compaction is by reducing the number of marginal cells at the serosa window (Fig. 2a)[12]. While it is in principle possible that leading cells are not excluded and converge to a multicellular rosette, such a rosette has not been observed during *Tribolium* serosa window closure[12,13,17]. Our cell tracking experiments showed that the initial number of approximately 75 leading cells progressively decreased to only 5–6 cells during final serosa closure (Fig. 2b, c) and that these cells originated from all around the periphery of the window (Fig. 2d and Supplementary Movie 3). Careful examination of individual cells at the leading edge in time-lapse recordings of embryos of the LifeAct-eGFP transgenic line revealed frequent rearrangement of cells resulting in cells leaving the serosal edge (Fig. 2e, f and Supplementary Movie 4). The leaving cells shrunk their leading edge facing the serosa window and elongated radially in the direction approximately orthogonal to the window (Supplementary Fig. 2A–C). Upon leaving the edge, the cells gradually relaxed to a hexagonal shape as they reintegrated into the bulk of the tissue (Supplementary Fig. 2D). Mapping of those behaviors onto the time-lapse cartographic projections revealed that the serosa was regionalized into two distinct territories. Dorsal cells, several cell diameters away from the edge, were hexagonally packed, isotropically stretched, and showed no significant neighbor exchanges. By contrast, ventral cells surrounding the serosa window were irregularly packed, showed anisotropically stretched

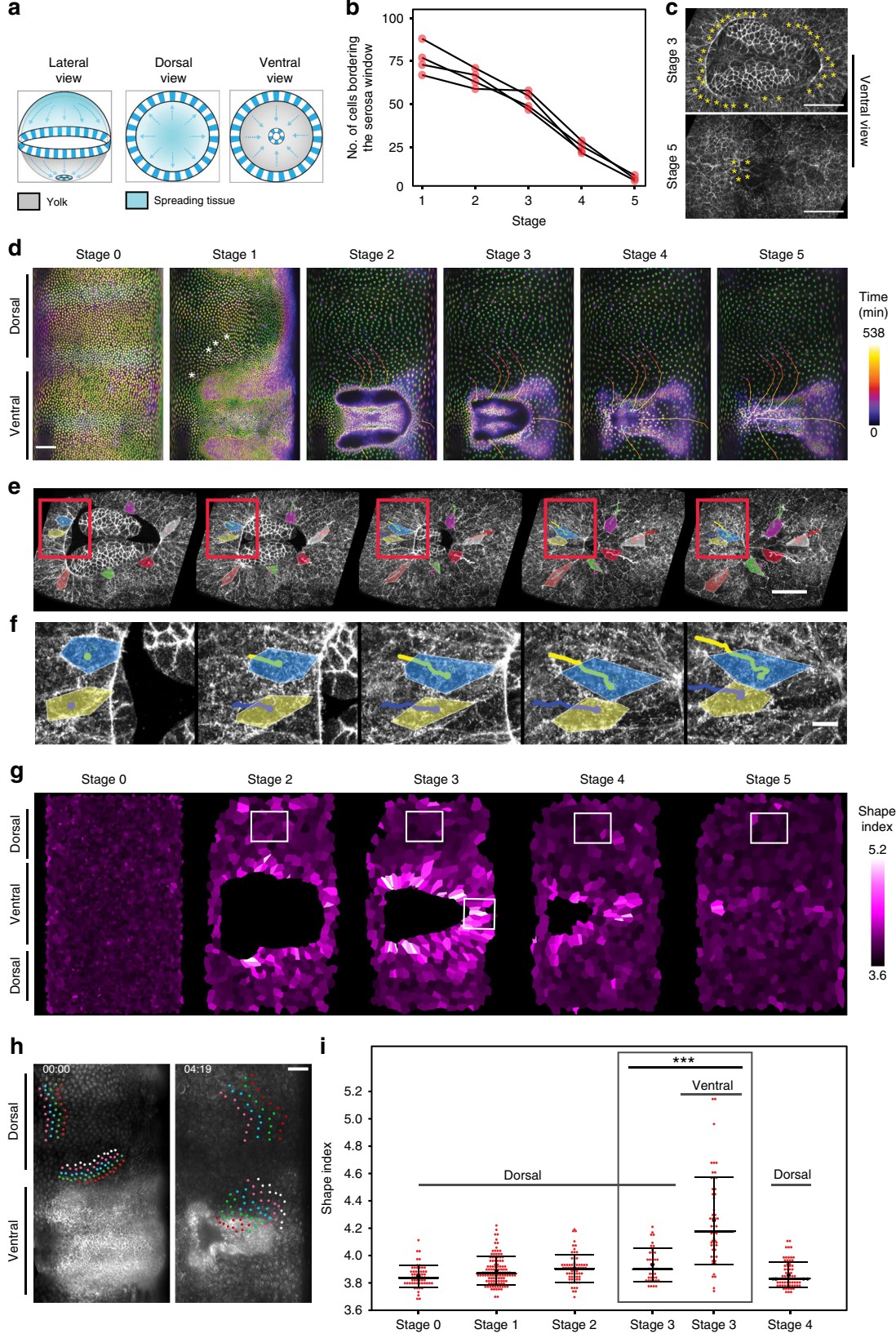

shapes (Supplementary Fig. 3A and Supplementary Movie 11), and frequently exchanged neighbors (Fig. 2h and Supplementary Movie 5).

Movement of cells past each other during neighbor exchange has been linked to increased tissue fluidity[18–21]. A useful theoretical framework to assess the behavior of the serosal tissue is the shape index analysis that infers solid-like or fluid-like tissue states from cell shapes in epithelia[22–24]. Based on the vertex model, the leading theoretical framework for studying the mechanical behavior of epithelial tissues[25], the theory predicts a

**Fig. 2 Cell behaviors at the serosal edge during window closure. a** Schematic illustration of the putative mechanism of closing serosa window by reducing the number of cells at the leading edge of the window over time. **b** Plot of the total number of cells at the extraembryonic–embryonic boundary during serosa window closure counted at the five reference stages ($N = 4$). **c** Confocal images highlighting the cells (yellow asterisks) forming the leading edge of the serosa window at Stage 3 (top) and Stage 5 (bottom). Scale bars are 10 μm. $N = 1$ (2 datasets available). **d** Cartographic projections of an embryo expressing Histone-eGFP imaged with multi-view SPIM. Progressively deeper layers of the projections are color coded to distinguish between superficial (green) and internal (magenta) nuclei. The nuclei participating in closing of the serosa window were back-tracked to the uniform blastoderm stage to reveal their spatial origin. Tracks are color coded by time as indicated by the color scale. Scale bar is approximately 100 μm. $N = 1$ (3 datasets available). **e** Frames from a confocal recording of serosa window closure in embryos expressing LifeAct-eGFP. Selected tracked cells at the leading edge of the serosa window are outlined and colored to show that they shrink their serosa-window-facing edges and planarly intercalate into the serosal epithelium. Scale bar is 50 μm. $N = 1$ (2 datasets available). **f** Close-ups of the cells inside the red boxes in **e**. Scale bar is 10 μm. **g** Segmented cartographic projections as in Fig. 1g with serosal cells color coded according to their shape index values. White boxes indicate the approximate regions from which cells sampled in confocal datasets were quantified in **i**. $N = 1$. **h** Cartographic projections of an embryo injected with *LifeAct-eGFP* mRNA shown at the beginning (left) and toward the end (right) of serosa window closure. Indicated rows of cells were tracked over time and color coded to visualize the difference in the extent of neighbor exchange between the dorsal cells and ventral cells close to the leading edge of the serosa. Scale bar is approximately 100 μm. $N = 1$. **i** Distributions of shape indices of segmented cells in Stages 0–4 in transgenic LifeAct-eGFP embryos imaged with confocal microscopy. Numbers of cells and embryos are the same as in Fig. 1f.

critical value of shape index $p = 3.81$ marking the transition from a solid-like ($p < 3.81$) to a fluid-like behavior ($p > 3.81$ but see also ref. [26] and below). Our results showed that at Stage 3 ventral cells had on average a high shape index $p$ of 4.25 characteristic of fluid-like tissues, unlike dorsal cells that had a significantly lower $p$ value of 3.93. The changing values of the shape index suggest a gradient of tissue properties along the dorsal–ventral axis of the embryo, where the ventral region is much more fluid-like compared to the dorsal region (Fig. 2g, i and Supplementary Movie 11). These results raised the hypothesis that during serosa epiboly the tissue in the vicinity of the window undergoes a solid-to-fluid structural transition (fluidization) that unjams the tissue and enables seamless closure.

**Serosa shows distinct mechanical properties along the dorso-ventral axis.** We next asked what the mechanical function of the ventral serosa fluidization could be. If the dorsal serosa behaves as a solid-like material, we expect that while being pulled over the egg it would increasingly build up tension. This rising tension would make it increasingly more difficult to further close the serosa window. The function of the ventral cell rearrangement in the proximity of the serosa window could then be in releasing this tension to facilitate closure. Consequently, we would predict a difference in tissue tension between dorsal and ventral serosa. To test this, we performed laser ablations inflicting large incisions across 3–4 cells at different reference stages and positions and determined the recoil velocities of the bordering tissue immediately after the cut[27] (Fig. 3a, b). The cuts were oriented perpendicular to the axis along which the serosal cells were stretched. Our results showed that the tissue recoil velocity in the dorsal side increased progressively as the serosa expanded posteriorly and ventrally around the posterior pole and plateaued after the serosa window formed (Fig. 3c). Intriguingly, when we performed incisions at the ventral side of the serosa at a stage where this tissue exhibits cell rearrangements (Stage 3), the recoil velocities were significantly lower compared to the dorsal side (Fig. 3d). Because the tissue recoil velocity depends on both the tension in the tissue and the material properties of the tissue, we further analyzed the time-dependent decay of the tissue recoil[28]. Such an analysis can discern between fluid-like or solid-like recoil patterns of the severed tissue. Our data suggested that the ventral tissue exhibits more fluid-like behavior than the dorsal tissue (Supplementary Fig. 4 and "Methods"). Therefore, the laser cutting experiments support the view that the dorsal tissue behaves like an elastic solid. Assuming that its properties do not change dramatically over the course of serosa expansion, the tension in this part of the serosa increases as the tissue gets stretched. This

notion is further corroborated by our observations that the intact cells neighboring an ablation site responded to the release in tissue tension post-ablation by immediately decreasing their apical areas by one-third (Supplementary Fig. 5). In contrast to the dorsal region, the ventral portion of the serosa is transitioning from a solid-like to a fluid-like state, which is connected to the local reduction of tissue tension.

While the recoil pattern after laser ablation supports the hypothesis of ventral tissue fluidization suggested by the shape index analysis, it has been recently shown that the relationship between shape index and tissue fluidity is non-linear when the tissue is under tension[26]. Since we obtained from laser ablations evidence that the *Tribolium* serosa exhibits a spatially inhomogeneous tension profile, we applied this extended theoretical framework. Moreover, we observed that the cells close to the window are strongly elongated in direction radial to the window (Fig. 2e, f and Supplementary Fig. 2) which could indicate local anisotropy in the tension profile. Therefore, we calculated a local cell alignment factor $Q$ across the serosal tissue as a proxy measure of local tissue tension anisotropy (see "Methods")[29]. The theory predicts that for a given value of $Q$ the shape index $p$ needs to exceed an adjusted threshold value in order for the tissue to be fluid like. For each local value of $Q$ across the cartographic maps, we plotted the difference between the actual shape index value ($p$) of the cell and the local threshold signifying solid-to-fluid transition (Fig. 3e, f and Supplementary Movie 11). This analysis revealed that, also when taking tissue tension anisotropy into consideration, the ventral cells lining the rim of the serosa window exhibited a distinct fluid-like state during closure, in stark contrast with the rest of the epithelium exhibiting a solid-like state. Therefore, both experimental and theoretical evidence support the local fluidization of the ventral-most serosal tissue.

**Ventral serosa fluidization is mediated by an actomyosin cable.** We next asked what induces the local tissue fluidization. Recent live imaging studies of *Tribolium* gastrulation suggested that an accumulation of actin, resembling a cable, emerges at the leading edge of the serosa[13,30]. To test whether this accumulation indeed represents a contractile actomyosin cable[31,32], we imaged the distribution of non-muscle myosin II (hereafter referred to as myosin) in gastrulating embryos injected with the *Tribolium* myosin regulatory light chain (*Tc-sqh*) mRNA fused to *eGFP* (*Tc-sqh-eGFP*). During epiboly, myosin accumulated at the boundary between the serosa and the embryonic primordium (Fig. 4a, b, Supplementary Fig. 6A–C, and Supplementary Movie 6). Acto-myosin enrichment at the serosa–embryonic boundary initiated shortly after epiboly started and became more pronounced as the

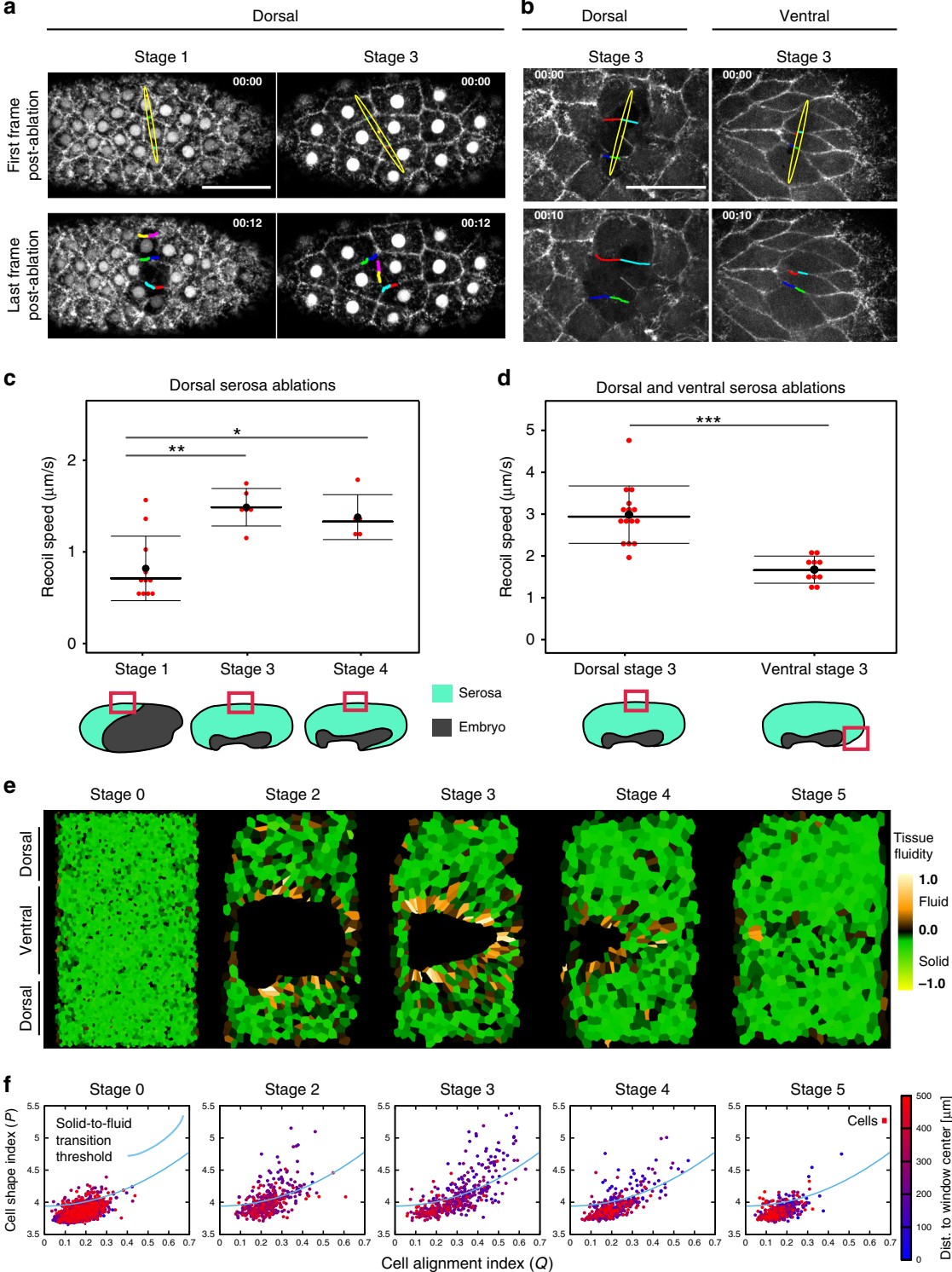

boundary stretched around the posterior pole. It peaked during serosa window closure and at this stage appeared as a contiguous supra-cellular cable (Fig. 4b and Supplementary Fig. 6A–C). The actomyosin cable lined the rim of the serosa window and underwent shape transformations from triangular to circular during closure (Fig. 4c and Supplementary Figs. 6B and 7A). The cable formed at the boundary of the serosa–embryonic (pro-spective amnion) region and the serosal cells forming the cable often appeared bent inwards over the window rim (Supplementary Fig. 7). By segmenting and measuring the length of the cable

based on LifeAct-eGFP or Tc-sqh-eGFP enrichment, we found that the cable first increased its length until the serosa–embryonic boundary reached the posterior pole and then decreased in length to zero during window closure (Supplementary Fig. 6D). As the cable shrank, the total myosin intensity normalized by cable length stayed the same or increased over time (Supplementary Fig. 6E). Laser cutting experiments of individual cell edges con-tributing to the actomyosin cable revealed that the cable was under tension and that this tension increased over time (Fig. 4d–f and Supplementary Movie 7). If the cable acted as a contiguous

**Fig. 3 Tension landscape in the expanding serosa. a** Tissue laser ablations in the dorsal serosa at different reference stages using a two-photon laser ablation set-up. Images show the serosal tissue before (top) and after (bottom) laser ablation in Stage 1 and Stage 3 embryos expressing LifeAct-eGFP and EFA-nGFP. Ablations were oriented perpendicular to the anterior–posterior axis of the embryo and yellow ellipses show the extent of the cut. The colored lines highlight the displacement of the severed cell edges. Time stamps are mm:ss. Scale bar is 50 μm. Stage 1 $N = 1$, Stage 3 $N = 1$. **b** Tissue laser ablations in the dorsal and ventral regions of the serosa using a UV laser ablation set-up. Images show the serosal tissue before (top) and after (bottom) laser ablation in Stage 3 embryos expressing LifeAct-eGFP. Ablations were oriented perpendicular to the anterior–posterior axis of the embryo. The position of the cuts in **a** and **b** are indicated with red boxes in the embryo illustrations below **c**, **d** yellow ellipses show the extent of the cut. Annotations are as in **a**. Time stamps are mm:ss. Scale bar is 50 μm. Dorsal Stage 3 $N = 1$, Ventral Stage 3 $N = 1$. **c** Graph showing the recoil velocities after laser ablations using a two-photon laser in Stage 1, 3, and 4 embryos. Each dot represents one cut in one embryo inflicted in the dorsal serosal region indicated by the red boxes in the reference illustrations below the graph. The number of embryos ($N$) sampled at different stages were as follows: Stage 1 $N = 11$, Stage 3 $N = 6$, Stage 4 $N = 5$. **d** Graph showing the recoil velocities after laser ablations using a UV laser of serosal cells in Stage 3 embryos. Each dot represents one cut in one embryo inflicted in the dorsal or ventral serosal region indicated by the red boxes in the reference illustrations below the graph. The number of embryos ($N$) sampled were as follows: Dorsal $N = 15$, Ventral $N = 10$. Distributions were compared using Welch's unpaired $t$ test. **e** Segmented cartographic projections as in Figs. 1g and 2g with serosal cells color coded according to their tissue fluidity values measured by subtracting the local solid-to-fluid transition shape index threshold (blue curve in **f**) from the cell shape index for each segmented cell (see "Methods" section "Shape index analysis"). Positive values indicate fluid-like and negative values solid-like properties. **f** Scatter plots of cell shape alignment index (x-axis) and shape index (y-axis) values of individual cells in the maps shown in **e**. The cells are color coded according to their distance from the center of the serosa window. The blue line indicates theoretically predicted threshold value of shape index signifying solid-to-fluid structural transition. Points below the line indicate solid-like cells and points above the line fluid-like cells (see "Methods" section "Shape index analysis").

contractile ring, one would expect global loss of tension after a cut. Instead, when we inflicted successive laser cuts at different positions of the same cable, the recoil velocities were comparable (Fig. 4g). This indicated that individual cells of the cable contract their myosin-loaded edges independently and implied that the serosa window edge acts as a chain of independently contractile units. Moreover, in a transgenic *Tribolium* line expressing Tc-sqh-eGFP from a ubiquitous promoter (see "Methods"), the myosin distribution around the cable circumference showed strong heterogeneity, with some cells exhibiting higher and other cells exhibiting lower myosin accumulation. Cells with more myosin contracted their cable-forming edges and were evicted from the leading edge of the serosa earlier than cells with lower levels of myosin (Fig. 4h, i and Supplementary Movie 8). Since the myosin intensity correlates with the cell-leaving behavior, we hypothesize that differential line tension along the cable circumference drives the eviction of the cells from the cable and the resulting cell rearrangements lead to tissue fluidization and eventual closure of the epithelial gap (Fig. 4j).

**Abolishing the cable halts ventral serosa fluidization and closure.** Such a model predicts that, in the absence of the actomyosin cable, the serosa window would fail to close ventrally. A previous study suggested that the juxtaposition of normally proportioned extraembryonic (serosa) and embryonic (amnion and germband) rudiments in the differentiated *Tribolium* blastoderm is required for proper emergence and constriction of the actomyosin cable at the extraembryonic/embryonic boundary[13]. Furthermore, it has been demonstrated that the transcription factor-encoding *zerknüllt-1* gene (*Tc-zen1*) has an early function in specifying serosal cell fate and that RNAi knock-down of *Tc-zen1* results in serosa-less embryos that are not covered by extraembryonic membranes ventrally[11]. Based on this evidence, we hypothesized that *Tc-zen1^RNAi* embryos would be lacking the actomyosin cable. Although *Tc-zen1* knock-down is expected to impact multiple cellular properties in the anterior blastoderm, where cells are transformed from serosal into embryonic (most likely amniotic) fate, *Tc-zen1^RNAi* embryos exhibit a very specific early morphogenetic defect without significantly compromising the morphology and viability of late embryos[11]. Live imaging of transgenic embryos expressing LifeAct-eGFP obtained after knock-down of *Tc-zen1* revealed indeed the absence of the actomyosin cable (Fig. 5a, b and Supplementary Movie 9). While such *Tc-zen1^RNAi*

embryos started the contraction and folding of the embryonic primordium as wild-type embryos, the epibolic movement halted and a ventral serosa window failed to form and close (Fig. 5a, b, e and Supplementary Movie 10). Compared to wild type, the dorsal spreading cells in *Tc-zen1^RNAi* embryos became larger, presumably due to their lower number (Fig. 5c). The cells on the ventral leading edge, however, were much smaller (Fig. 5d, f), did not elongate anisotropically (Fig. 5b and Supplementary Fig. 3B), did not exchange neighbors, and were not evicted from the leading edge. Finally, although the shape index of the dorsal cells in *Tc-zen1^RNAi* embryos was comparable to wild type (Fig. 5g, h), the ventral region showed a significantly lower shape index compared to wild type (Fig. 5g, i) with less pronounced regionalization around the serosal window (Fig. 5g, j). We propose that one reason why the serosa window fails to close in the absence of the actomyosin cable is because tissue fluidization fails to occur and the epithelial tissue cannot remodel to close its gap.

## Discussion

The epibolic expansion of the *Tribolium* serosa to envelop the entire egg surface is a dynamic morphogenetic process constrained by the ellipsoidal geometry of the egg and the mechanical properties of the tissue. Our data suggest that the regionalized tissue fluidization at its leading edge solves the geometrical and mechanical problems associated with serosa epiboly. First, it addresses the geometric constraints necessitating both the expansion and regional compaction of the tissue to close the gap. While the bulk of the tissue expands in a manner similar to an elastic solid material, the fluid-like ventral region remodels, halts the increase in cell area, and therefore can remain compact. Second, in the absence of cell divisions, which have been implicated as a stress-release mechanism in fish[4,33], local cell rearrangements induced by actomyosin contractility at the leading edge release the mechanical stress in the non-proliferative serosal sheet and maintain epithelial integrity during closure.

While here we focused on the gradient of serosa properties along the dorsal ventral axis, the pulling of the epithelial sheet over an ovoidal shape is an inherently 3D process and thus the tissue likely experiences stresses in many directions. More systematic probing of the mechanical properties of gastrulating *Tribolium* embryos will be required to understand the source of the forces acting on the serosa. We expect that the condensation of the adjoining embryonic primordium[13,14], together with

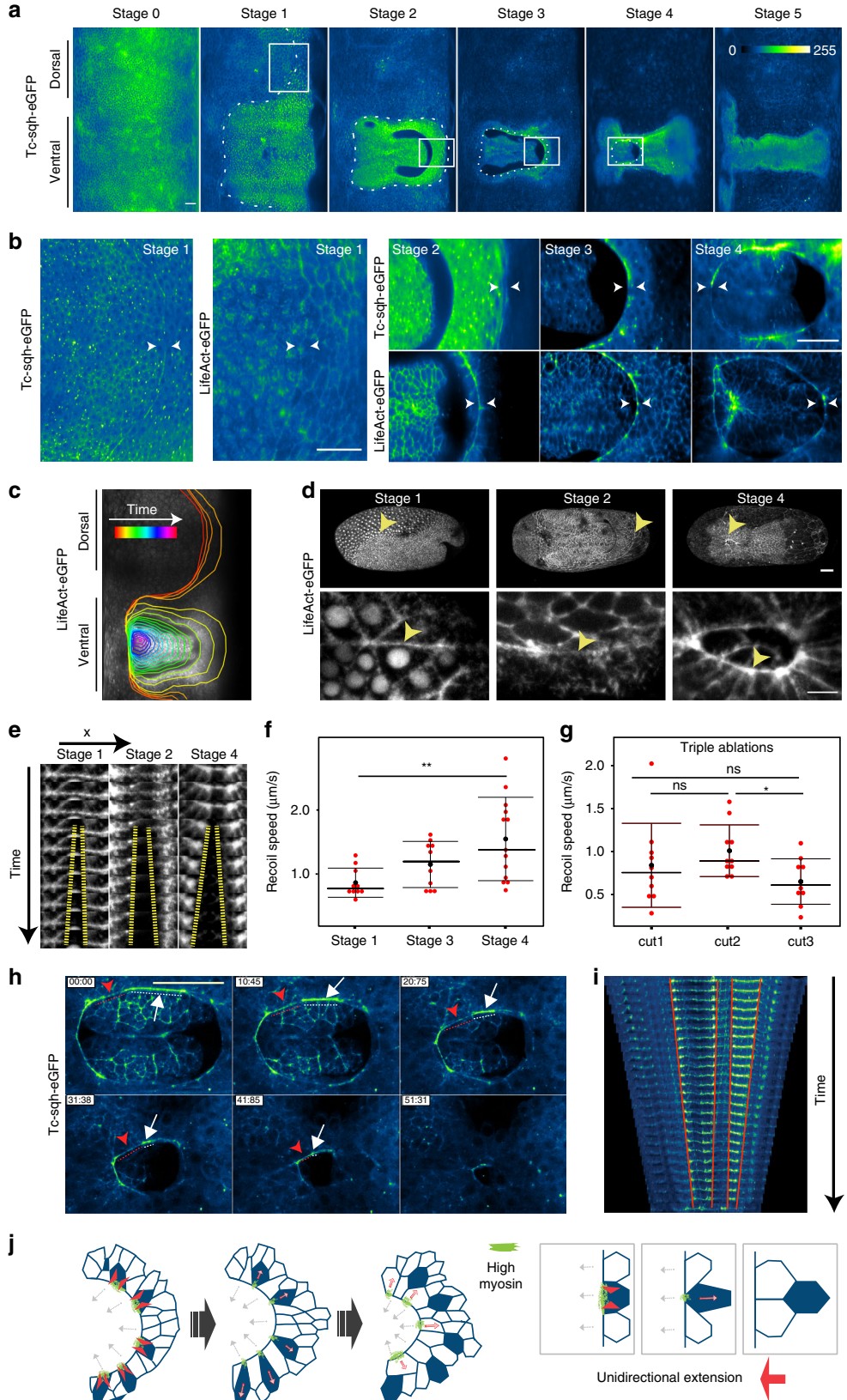

additional forces generated by the attachment of the blastoderm to the vitelline envelope[30], the yolk[13], active crawling of the serosa on the vitelline envelope[12], and regulated changes in the shape and stiffness of the serosal cells[17,34,35], may also contribute to the serosal epiboly.

Independent of the forces involved in serosa expansion, our results suggest that the serosa closure is mediated by a heterogeneous actomyosin cable operating at the single-cell level to exclude marginal cells individually from the serosa window. The order in which cells are evicted correlates with local myosin

**Fig. 4 Cell eviction by a heterogeneous actomyosin cable at the serosal edge during window closure. a** Cartographic projections of *Tribolium* embryos injected with *Tc-sqh-eGFP* mRNA and imaged with multi-view SPIM. The accumulation of myosin at the extraembryonic–embryonic boundary is highlighted by the dotted line as it emerges around the egg circumference (Stage 1) and then during its progressive constriction on the ventral side of the embryo (Stages 2-5). Scale bar is approximately 50 µm. Intensity in all panels is color coded with the Green (high)–Blue (low) LUT. *N* = 1 (3 datasets available). **b** Insets show zoomed-in images of Tc-sqh-eGFP localization in the regions marked by white boxes in **a** and similar images from cartographic projections of an embryo injected with *LifeAct-eGFP* mRNA and imaged with multi-view SPIM (*N* = 1, 2 datasets available). Actomyosin enrichment is shown between the arrowheads. **c** The shape of the actomyosin cable in a map-projected LifeAct-eGFP SPIM recording is outlined over time as it emerges dorsally and closes on the ventral side of the embryo. The color of the outline corresponds to the time stamp of the frame from which it was traced. *N* = 1. **d** Maximum intensity projections of confocal stacks of embryos expressing LifeAct-eGFP from three different developmental stages. Arrowheads point to the regions of the cable that was ablated. Stage 1 images are lateral views and Stage 2 and 4 images ventral views. Bottom row shows close-ups of areas marked by arrows in the top row. Scale bars are 50 µm in top panels and 10 µm in bottom panels. Stage 1 *N* = 1, Stage 2 *N* = 1, Stage 4 *N* = 1. **e** Kymograph of the recoiling membrane edges (yellow hyphen) after laser ablation of the cells forming the actomyosin cable at the leading edge of the serosa window. Stage 1 *N* = 1, Stage 2 *N* = 1, Stage 4 *N* = 1. **f** The distributions of recoil velocities after ablation of the cable-forming cells at different stages. The number of embryos (*N*) sampled at different stages were as follows: Stage 1 *N* = 10, Stage 3 *N* = 10, Stage 4 *N* = 13. Distributions were compared using Welch's unpaired *t* tests. **g** The distributions of recoil velocities after three successive laser ablations of three distinct cable-forming cell edges in a single cable at Stage 4. The number of embryos (*N*) and successive cuts (*n*) were as follows: *N* = 10, cut1 *n* = 10, cut2 *n* = 10, cut3 *n* = 10. Distributions were compared using Welch's unpaired *t* tests. **h** Images from a time-lapse confocal recording of a Tc-sqh-eGFP transgenic embryo. Myosin localization at the cable varies between different cable-forming cells. A cell with high myosin accumulation is labeled with white arrow and its extent is highlighted with white dotted line. A cell with low myosin is labeled similarly but in red. Time stamps are mm:ss. Scale bar is 10 µm. *N* = 1 (5 datasets available). **i** Kymograph of myosin cable shown in **h**. The cable was segmented manually and straightened computationally in Fiji. **j** Illustration shows the differential contraction of the serosa-window-facing cell edges depending on the amount of myosin. This leads to T1 transitions in the serosa (right). As a result, the leading edge of serosa extends unidirectionally (gray arrows) and at the same time undergoes structural rearrangement. Green color depicts the myosin enriched in the contracting cells (red arrowheads).

accumulation at each cable-forming edge. This is consistent with previous findings that myosin intensity correlates with tension in wound-healing cables[36,37]. Furthermore, it has been suggested that a non-uniform stepwise contractility of individual edges is necessary for efficient epithelial closure during wound healing in *Drosophila* embryos and neural tube closure in chordates[38,39]. This kind of sequential contraction is likely operating during window closure to dissipate serosal resistance. Last but not the least, a recent study proposed that tissue fluidization is required for seamless wound healing in damaged *Drosophila* imaginal discs[18]. Similar to the actomyosin cable of the *Tribolium* serosa window, the cable that assembles at the leading edge of the wound evicts cells from the wound periphery and promotes cell intercalation resulting in tissue fluidization and acceleration of epithelial gap closure. All these striking similarities point toward a general morphogenetic function of actomyosin cables in shaping and repairing epithelia by local tissue fluidization.

## Methods

***Tribolium* rearing and stocks**. *T. castaneum* stocks were kept at 32 °C and 70% relative humidity on whole-grain or white flour supplemented with yeast powder according to standard procedures[40]. All mRNA injections were performed into embryos of the *vermilion^white* strain. The following transgenic lines were used for live imaging: (i) *EFA-nGFP*, ubiquitously expressing a nuclear-localized GFP reporter[41] (kindly provided by Michalis Averof's laboratory); (ii) *αTub-H2A-eGFP*, ubiquitously expressing a nuclear eGFP reporter (kindly provided by Peter Kitzmann from Gregor Bucher's laboratory); (iii) *EFA-Gap43-YFP-2A-Histone-RFP*, ubiquitously expressing both a membrane YFP and a nuclear RFP reporter (kindly provided by Johannes Schinko and Anna Gilles from Michalis Averof's laboratory); (iv) *αTub-LifeAct-eGFP*, ubiquitously labeling filamentous actin with eGFP[16] (kindly provided by the Van der Zee laboratory); and (v) *αTub-Tc-sqh-eGFP*, ubiquitously labeling the *Tribolium* non-muscle myosin II through its regulatory light chain. The predicted *Tribolium spaghetti squash* gene (*Tc-sqh*) encoding the non-muscle myosin II regulatory light chain was identified by BLAST analysis against the *Tribolium* genome[42] and shares 93% similarity with the *Drosophila melanogaster sqh* gene. The *Tc-sqh* open reading frame was amplified from cDNA and cloned in-frame with *eGFP* downstream of the *Tribolium α-Tubulin1* promoter[43] in a *piggyBac* transgenesis vector, kindly provided by Peter Kitzmann and Gregor Bucher. The pBac-αTub-Tc-sqh-eGFP vector was injected into the *Tribolium vermilion^white* strain together with a helper plasmid expressing the *piggyBac* transposase and eight transgenic lines were established by the TriGenes gUG service (https://trigenes.com). Experiments were conducted using two selected transgenic lines exhibiting ubiquitous and uniform expression of Tc-sqh-eGFP.

Overview of genotypes and constructs used in the study is provided in Supplementary Table 1.

**RNA injections**. Actin and myosin dynamics were visualized in *vermilion^white* embryos injected with in vitro transcribed capped mRNAs encoding *LifeAct-eGFP* or *Tc-sqh-eGFP* that were synthesized from linearized plasmid templates pT7-LifeAct-eGFP and pCS2+-Tc-sqh-eGFP, respectively[13,30]. For the RNAi knock-down experiments of *Tc-zen1*, the double-stranded RNA (dsRNA) against the *Tribolium zerknüllt-1* gene (TC000921) was synthesized with primers optimized for gene specificity (a 203-bp amplicon outside of the conserved homeobox region)[44]. The mRNAs and the dsRNA were each injected at a concentration of 1 mg/ml. Eggs from the *vermilion^white* strain were collected for 2 h at 30 °C, aged for another hour at 30 °C and dechorionated in 16% commercial Klorix bleach for 1–2 min. Dechorionated pre-blastoderm embryos were mounted on a 1% agar bed and were micro-injected in air through their anterior pole under a brightfield upright microscope, as previously described[13,40] (Supplementary Fig. 1A). Injected eggs were incubated in humid chambers at 30 °C for ~2 h and the most homogeneously labeled and bright embryos were selected for imaging. For parental knock-down of *Tc-zen1* by RNAi, dsRNA was injected into the abdomen of female pupae collected from the *αTub-LifeAct-eGFP* transgenic line[16]. Injected adult females were crossed to males from the same line and their eggs were collected for imaging. For the embryonic RNAi knock-down experiments, 0–3 h-old embryos were co-injected with mRNA encoding fluorescent reporters and dsRNA against *Tc-zen1*.

**Live imaging with confocal and light-sheet microscopy**. Point scanning confocal live imaging was carried out at 25 °C or 30 °C on an inverted Zeiss LSM 780 system equipped with a temperature-controlled incubator. Embryos were mounted in 1% agarose in glass bottom petri dishes and covered in water. Embryos were scanned with a Zeiss ×25/0.8 NA Plan-Apochromat multi-immersion objective or a Zeiss ×40/1.2 NA C-Apochromat water-dipping objective with pixel sizes ranging between 0.2 and 0.55 µm, a *z*-step of 2 µm, and a temporal resolution of 5 min. Multi-view light-sheet imaging, referred to as selective plane illumination microscopy (SPIM) or light-sheet microscopy, was carried out on a Zeiss Lightsheet Z.1 microscope equipped with a ×20/1.0 NA Plan Apochromat water-immersion detection objective and two ×10/0.2 NA dry illumination objectives. Embryos were embedded in glass capillaries in 1% low melting agarose dissolved in 1× phosphate-buffered saline together with fluorescent beads, as previously described[45,46]. For each embryo, z-stacks were acquired from different views with the following voxel sizes: Figs. 1c, g, 2d, g, 3e, 4b (3 views every 120° with voxel size 0.381 µm × 0.381 µm × 2.0 µm), Fig. 1d (3 views every 120° with voxel size 0.33 µm × 0.33 µm × 2.0 µm), Fig. 2h, 4a, b, 5e–g, j (5 views every 72° with voxel size 0.381 µm × 0.381 µm × 2.0 µm, Fig. 4b Tc-sqh-eGFP Stage 1 (4 views every 60° with voxel size 0.381 µm × 0.381 µm × 2.0 µm. The starting point in the time stamps used for all experiments was the last (12th) round of synchronous nuclear divisions, which precedes the formation of the uniform blastoderm and all subsequent morphogenetic events[12]. Parameters for all live imaging experiments are summarized in Supplementary Table 1.

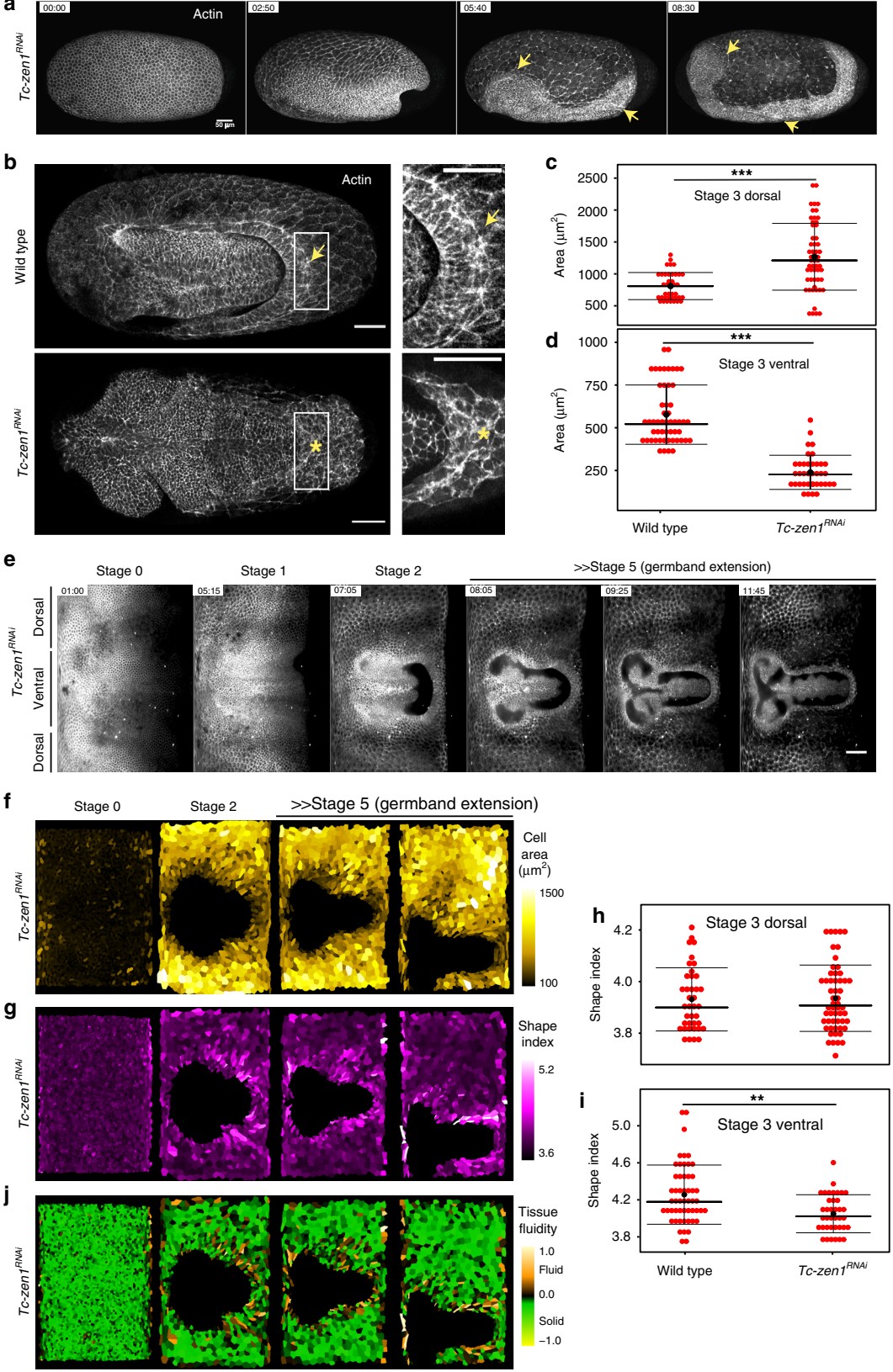

**Laser ablations**. Laser ablations were performed either on an inverted Zeiss LSM 780 NLO with a ×40/1.2 NA water-dipping objective using an 800 nm pulsed infrared laser or on a customized spinning disc confocal unit with a ×63 water-dipping objective using an ultraviolet laser microdissection apparatus similar to the one described in Grill et al.[47]. On the first system, three planes with 1–2 μm z-spacing were imaged every 1.6 s (Fig. 4f), 2.5 s (Fig. 4g), or 2.6 s (Fig. 3c) and the

cut was performed in the middle plane, while on the latter system a single plane was recorded every 0.5 s (Fig. 3d). Tissue cuts were about 12 μm long spanning 3–4 cell diameters, while ablations of single edges were about 5 μm long. The recoil velocity of ablated edges was measured between six post-cut time frames using the manual tracking plugin in Fiji. For each cut, two to three independent tracks of the recoiling tissue edges were averaged. The recoil velocity was estimated using

**Fig. 5 Cell and tissue dynamics in *Tc-zen1* knock-down embryos. a** Maximum intensity projections of a parental *Tc-zen1*[RNAi] embryo labeled with LifeAct-eGFP and imaged with confocal microscopy. Arrows point to the open serosa window in the head and the posterior region. Time stamps are hh:mm. Scale bar is 50 μm. $N = 1$ (2 datasets available). **b** Selected maximum intensity projection images from wild-type (top) and parental *Tc-zen1*[RNAi] (bottom) embryos at Stage 3. Insets show the cable in wild-type embryos (arrow) and absence of the cable in the knock-down (*). Scale bars are 50 μm. Wild type $N = 1$, *Tc-zen1*[RNAi] $N = 1$. **c** Distribution of apical cell areas in wild-type (left) and embryonic *Tc-zen1*[RNAi] (right) embryos sampled from confocal datasets in the dorsal serosa at Stage 3. The number of cells ($n$) and embryos ($N$) sampled are $n = 39$ and $N = 6$ wild-type embryos and $n = 55$ and $N = 7$ *Tc-zen1*[RNAi] embryos. **d** Distribution of apical cell areas in wild-type (left) and embryonic *Tc-zen1*[RNAi] (right) embryos sampled from confocal datasets in the ventral serosa at Stage 3. The number of embryos ($N$) and cells ($n$) sampled are $n = 52$ and $N = 7$ in wild-type and $n = 38$ and $N = 7$ in *Tc-zen1*[RNAi] embryos. **e** Cartographic projections of embryonic *Tc-zen1*[RNAi] embryo injected with *Gap43-eYFP* mRNA reconstructed from a multi-view SPIM recording. Time stamps are hh:mm. Scale bar is approximately 100 μm. $N = 1$. **f** Cartographic projections shown in **e** overlaid with outlines of serosal cells. Serosal cell in each projection were segmented automatically, curated manually, and color coded according to their apical cell area. $N = 1$. **g** Segmented cartographic projections as in **f** color coded according to the shape index of the segmented serosal cells. **h** Distribution of shape indices in wild-type and *Tc-zen1*[RNAi] embryos sampled from confocal datasets in the dorsal serosa at Stage 3. Numbers of cells and embryos are the same as in **c. i** Distribution of shape indices in wild-type and *Tc-zen1*[RNAi] embryos sampled from confocal datasets in the ventral serosa at Stage 3. Numbers of cells and embryos are the same as in **d. j** Segmented cartographic projections as in **f**, **g** color coded according to the tissue fluidity values of the segmented serosal cells (see "Methods" section "Shape index analysis").

standard procedure of measuring the total distance traveled by the two recoiling edges between frames 1 and 2 post-cutting and dividing by acquisition frame rate[48]. The observed differences in absolute recoil velocities between the ultraviolet and two-photon laser ablation set-ups used in this study have been observed before and are thought to reflect difference in the extent of disruption of the actomyosin network[36,49]. Keeping this in mind, we only compared recoil velocities generated with the same laser ablation set-up, as shown in Fig. 3c, d.

In an attempt to estimate tissue material properties from recoil dynamics after laser ablation, power-law analysis was performed as previously described[28]. Briefly, a kymograph perpendicular to the laser cuts was obtained with the Multi-Kymograph plugin in Fiji [https://imagej.net/Multi_Kymograph]. The recoil velocity was quantified during the first 10 s after the ablation and fitted by a power-law $D(t - t_0)^\alpha$, where $t_0$ is the time of the ablation and $D$ and $\alpha$ are fitting parameters. $D$ correlates with the recoil velocities of the edges, while $\alpha$ gives an indication of the mechanical properties of the tissue and can acquire values from 0 to 1. A tissue with a value of $\alpha$ closer to 0 shows properties of an elastic solid, while a value closer to 1 indicates viscous fluid properties.

**Image processing**. The multi-view light-sheet datasets were registered and fused using Fiji plugins as previously described[50–52]. The four-dimensional (3D + time) fused datasets were converted into 3D (2D + time) time-lapse maps by making cylindrical projections using the ImSAnE software[15]. Cells were segmented using a deep learning-based approach called StarDist, which is capable of learning morphological priors[53]. Different neural networks were trained for different markers (membrane and actin labels). The training data were obtained by generating realistic looking synthetic microscopic images of *Tribolium* using Generative Adversarial Networks. The generated synthetic data were evaluated visually against the real microscopic data to ensure textural and morphological consistency between the two. After training StarDist networks on such synthetic data, they were applied to the real microscopic images, and the predictions were manually curated with the Labkit plugin in Fiji (http://sites.imagej.net/Labkit/) to fix any segmentation mistakes. After cartographic projections, some cells on the edges of the maps were necessarily cut in order to unfurl the 3D embryo to 2D. Those incomplete cells were excluded from analysis. Distortions that are inherent to the mapping of curved surfaces onto a plane were corrected with custom Fiji plugins (available on the "Tomancak lab" Fiji Update site) thereby allowing the measurement of quantities like size, circularity, shape factor, density, velocity, and the local cell alignment (see below). Consequently, the scale bars in map projections are only approximate and reflect accurately the sizes only in the middle portions of the maps. Nuclei in the depth color-coded cartographic projections were tracked using MaMuT[45] and Mastodon (both available via Fiji Update sites).

**Shape index analysis**. Shape index was calculated for each segmented cell in the 2D cartographic projections as $p = P/\sqrt{(A)}$, where $P$ is the cell perimeter and $A$ is the cross-sectional area[22]. The measurements were distortion-corrected using the above-mentioned Fiji plugins and plotted onto the segmented cartographic projection as a color map. The local cell shape alignment index $Q$ was calculated as described recently[26]. Briefly, cells in each map projection were converted into a triangular mesh connecting the centers of all adjacent segmented cells, i.e., where three (or more) cells touch, a triangle (or a triangle fan) with vertices coinciding with centers of adjacent cells was formed. For every triangle, a degree $q$ of deviation from equilateral triangle was computed[29]. For every cell, its shape alignment index $Q$ became a weighted average over $q$ from all triangles whose vertex coincides with this cell's center. Using this $Q$, an adjusted shape index threshold was determined as $p_{adj} = p_0 + 4bQ^2$ for $p_0 = 3.94$ and $b = 0.43$[26]. According to Wang et al.[26],

simulations of this threshold marks solid-to-fluid transition for a given anisotropy in the tissue (i.e., for a given value of cell shape alignment index $Q$). The tissue fluidity for a given cell was then calculated as a difference between its actual shape index $p$ and $p_{adj}$ for a given local value of $Q$. This difference was converted into a color code and displayed on each cartographic projection. Green color signifies solid-like local tissue properties ($p < p_{adj}$), brown color fluid-like local tissue properties ($p > p_{adj}$), and black color marks the vicinity to the theoretically predicted solid-to-fluid transition ($p = p_{adj}$).

**Reporting summary**. Further information on research design is available in the Nature Research Reporting Summary linked to this article.

## Data availability
The confocal imaging data and cartographic maps that support the findings of this study are available on Figshare under the public project "Regionalized tissue fluidization is required for epithelial gap closure during insect gastrulation" (https://figshare.com/projects/Regionalized_tissue_fluidization_by_an_actomyosin_cable_is_required_for_epithelial_gap_closure_during_insect_gastrulation/86741). Raw light-sheet microscopic data are available on the Image Data Resource (https://idr.openmicroscopy.org) under accession number idr0099 or from P.T. upon request. The Figshare and IDR data DOIs are listed in an Image Datafile. All statistics and $p$ values are reported in the Statistics Datafile. Source data are provided with this paper.

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

## Acknowledgements

We thank Thorsten Horn for teaching us various *Tribolium* techniques; Matthew A. Benton for kindly providing the pT7-LifeAct-eGFP plasmid and sharing embryonic injection protocol and discussions; Sebastian Streichan for optimizing ImSANE for *Tribolium* SPIM data and critical discussions; Alexander Dibrov for helping with tissue cartography cell segmentations; Matthias Merkel for providing code for tissue shape analysis; Christopher Schmied for optimizing Snakemake SPIM data analysis pipeline for our datasets; Michaela Burkon for helping with *Tribolium* stock keeping and performing parental RNAi experiments; the MPI-CBG Light Microscopy Facility for assistance with imaging experiments; Mette Handberg-Thorsager and Yu-Wen Hsieh for sharing plasmids and help with cloning; Ivana Viktorinova for generating the schematic drawings; Anna Gilles and Johannes Schinko (Averof laboratory), Peter Kitzmann (Bucher laboratory), and the van der Zee laboratory for sharing valuable transgenic lines; and Siegfried Roth for critical discussions. A.J. was supported by the DIGS-BB Fellow Award. A.P. was supported by HHMI and IMBB-FORTH intramural funds. P.T. and S.W.G. were supported by the German Federal Ministry of Research and Education (BMBF) under the code 031L0044. P.T. and V.U. were supported by the European Regional Development Fund in the IT4Innovations national super-computing center - path to exascale project, project number CZ.02.1.01/0.0/0.0/ 16_013/0001791 within the Operational Programme Research, Development and Education.

## Author contributions

A.J. designed the research, performed experiments, analyzed the data, and wrote the manuscript. V.U. produced image analysis software and analyzed data. A.M. contributed to data analysis, M.P. helped in segmenting data, L.G.P. helped in laser ablation experiments, S.M. contributed reagents and data and was involved in discussions, R.H. produced image analysis software and contributed to analysis workflow design, M.B.C. performed the power-law analysis, K.A.P. conducted RNAi parameter validation experiments and was involved in discussions, F.J. contributed to data segmentation, S.W.G. helped in interpreting laser ablation data and was involved in discussions, P.T. and A.P. conceived and oversaw the project and wrote the manuscript.

## Funding

## Competing interests

The authors declare no competing interests.
