## [Peer Review File · Nature Communications]

Reviewers' Comments:

Reviewer #1:

Remarks to the Author:

Using confocal and state of the art light sheet live imaging, the authors provide here a concise and comprehensive description of cellular events during early morphogenetic movements of the extra-embryonic serosa in the flour beetle *Tribolium*. With the help of laser scissors and RNA interference, they functionally probe the role of tissue tension and an actin/myosin cable at the growing/closing edge of the serosa to demonstrate intriguing similarities with other morphogenetic processes including wound healing or epiboly in vertebrates.

This is a very solid paper of high relevance beyond insect development that in my view is very well suited for NatComms. I have only minor corrections/questions as detailed below.

(1) One major finding of this work is that the actin/myosin cable is essential to close the serosa window, and that inhomogeneities in cable thickness function to determine which cells are evicted next from the growing edge. While I think this interpretation is very likely correct, there is a potential problem in their argument: they visualize the cable through injection of a capped RNA encoding Myosin-GFP. However, given cell volumes in the blastoderm stage (when cellularization occurs) seem not very uniform, it is conceivable that much of the cell-to-cell variability of Myosin-GFP at later stages could be due to variation in the number of RNA molecules taken up by these cells at an early stage, or be due to stochastic degradation of mRNA.

I wonder if the authors have considered staining fixed embryos using crossreacting antibodies (or maybe in situ hybridization) to verify that also the endogenous myosin2 is unevenly expressed in the cells of the leading edge.

(2) Their finding that tissue tension is higher dorsally than ventrally does fit their concept of tissue fluidity but raises the question how the mechanical force is generated that places tension upon the dorsal serosa. About this I have two questions: (2a) did they probe ventral tissue tension through cuts parallel to the cable as well as at 90° to the cable, and if so was there a difference?

(2b) if the force pulling at the dorsal serosa is not (alone) exerted by the ventral serosa (including the actin/myosin cable), where else is this pulling force coming from? Do they think the dorsal serosa cells actively "crawl" along the vitelline membrane?

(3) Legend Fig. 3C, D; line 551: why are dorsal recoil speeds in C (stage 3) only half of those in D (same stage and location!)?

List of minor issues:

Line 243: "Anna Giles" is misspelt (=> Gilles)

Legend Fig. 3E, F: maybe in the legend the authors should refer to the methods section where these indices are defined

Legend Fig. 4A and elsewhere throughout the paper and supplemental info: "... injected with Tc-sqh-eGFP", Tc-sqh-eGFP should be italic to indicate that capped in vitro transcribed RNA, not protein was injected.

Legend Fig.4C: the time stamp color code of the cable (blue => purple => red => yellow => green) does not reflect the color bar (blue => green => red)

Legend Fig. 4 I: "... shown in (I)" should be (H)?

Legend Fig. S1B: " bright dots surrounding embryo" => it is in Fig. S1C where these white dots

are visible

Legend Fig. S2A: remove "a"

Legend Fig. S2D: indicate, to which time point in C "zero" corresponds. Also, it seems that anisotropy has been normalized for each cell at time zero, this should be mentioned in the legend.

Fig. S3: is the anisotropy at the poles (left and right edges of projections) real or an artifact of cartographic projections?

Supplementary movie 4: some relevant cells have been color-highlighted to facilitate tracking. This in part obscures the original Life-Act-GFP signal, making it hard to actually see the original signal. I would like to see the original movie and suggest to also include in the supplemental material the unlabeled version of the movie.

Supplementary movie 11: it should be explained also in the legend, why there is a dorsal gap in the projections

(Martin Klingler)

Reviewer #2:

Remarks to the Author:

In this manuscript Jain and co-workers have identified a novel mechanism driving epithelial cell rearrangement during epiboly in tribolium. Using a combination of quantitative imaging approaches and laser cut experiments they found that during epiboly, serosa cells are compartmentalized into two distinct territories displaying distinct mechanical properties. A dorsal region displaying more solid and elastic characteristic (as clearly demonstrated by the calculated recoil velocities upon laser cut), and a more ventral region that is more fluid and less elastic. They found that this behavior is determined by the assembly of a heterogeneous contractile actomyosin at the leading edge of the serosa region that by contracting facilitates cell intercalation away from the margin. Although a similar mechanism has been recently described during wound healing in response to injury in other contexts, this study provides the first demonstration that such a mechanism underlies the regulation of a developmentally controlled morphogenetic process. In general, I find that the experiments are well conducted and properly controlled with detailed information and statistical analysis and I therefore support publication of this manuscript in Nature Comm. I have two suggestions which I think will help strengthen the conclusions of this study.

- 1) Given the focus of this manuscript on tissue fluidity, I think it would be important to perform "creep and recovery" measurements to directly measure differences in tissue fluidity either by pipette aspiration or by injection of magnetic beads (if at all possible in tribolium).
- 2) The authors should explain better why they hypothesized that removal of *zerknüllt-1* would result in loss of the assembly of the actomyosin cable. They should also discuss that while the results of this experiment support their conclusion, the interpretation of this particular mutant phenotype might be more complex. *Zerknüllt-1* is a transcription factor expressed in all serosa cells and could impact on additional cellular behaviors that were not the focus of this analysis. It is tempting to suggest additional optogenetic experiments to more precisely dissect the role of actomyosin cable contraction in controlling tissue fluidization. However, I do appreciate that these experiments are not feasible at this point in tribolium.

Reviewer #3:

Remarks to the Author:

Jain et al. report on the cellular changes and anisotropies that occur during amnioserosa formation/function in *Tribolium*. They apply good quantitative skills and analysis to measuring cell shapes and tissue movements as the amnioserosa expands across the *Tribolium* embryo, and analogize these movements to epiboly processes in vertebrate organisms. They further interrogate tissue tensions through laser cuts. I found the results (especially the heterogeneity and analysis of the actomyosin cable) interesting, and I think it is solid work, but I am not sure the novelty or insight of the results rises to the level of *Nature Communications*. I think the current work lacks a certain mechanistic depth and is somewhat descriptive (some of the findings are unsurprising, at least to me), but this is not to say that it isn't highly meritorious. I also found the fluidization by intercalary movements not especially novel (again, nice for this system), but not profoundly insightful. For me, this would make for a very solid publication in *Development* – however, this is an admittedly subjective criteria and I am happy to hear if other reviewers have different opinions. The manuscript is well-written, and statistics are (largely) done appropriately. More detailed comments are below:

1) There was one problematic statement, which can probably be addressed textually, but also brings up one of the central weaknesses of the manuscript (lines 184-187), "Since the myosin intensity correlates with the cell leaving behavior, we conclude that differential line tension along the cable circumference drives the eviction of the cells from the cable and the resulting cell rearrangements lead to tissue fluidization and eventual closure of the epithelial gap" – this was just a correlation, but the authors have written a statement of causation, which can't be done based on the data to this point of the manuscript (could be rephrased as a hypothesis). I think the authors are setting up the *zen* RNAi experiments in the following paragraph, which are also problematic. The *zen* RNAi is interpreted as actin-cable-less embryos, but this is really a drastic cell fate transformation of the amnioserosa. For me, this is too multifaceted a change to represent a clean testing of actomyosin cable function, and therefore a full testing of one of the main hypothesis of the manuscript is not performed. Could Y-27632 be introduced at various points to test Myosin function more acutely and specifically?

2) The authors are mostly good at this, but statistical tests and n numbers should be listed in the figure legend for every figure panel where appropriate (some panels were missing this information). Please justify that data is normal and appropriate for parametric analysis where this is used.

3) A little more detail on the cell shape and Q shape method is needed – it was hard in reading the manuscript to fully understand what was going on in this section (without doing more reading of the Methods and background papers (which I did to a limited degree)). I know this is based off of other labs' work, but I don't fully buy the applicability of these thresholds across epithelial systems and it seems that some validation in this system is needed.

Minor notes:

The writing really is nice, and the entire preparation of the manuscript is well done. The authors are to be complimented on a nice piece of work.

Reviewer #4:

Enabled by recent advances in light sheet imaging and analysis tools the authors provide an elegant 3D description of serosal epiboly in the short germ band model insect *Tribolium castaneum*. In toto imaging utilizes fluorescent-protein tagged nuclei and the authors use additional probes to reveal actomyosin localization at the margin of the serosa as it encloses the developing embryo. To explain these movements and the physical mechanics of closure the authors adopt recent foam-based theory. This theory posits that one can surmise the mechanical properties based exclusively on the perimeter to area ratio of cells. Cells with longer perimeters than an equivalent isodiametric cell are considered "fluid". This theory is well suited for 2D conditions that confluent cultured epithelial cells inhabit, when cell sheets are most like 2D foams, but quickly fail when cell sheets inhabit more dynamic environments, such as embryos where cells may be patterned with distinct identities, different mechanical properties, and different levels of mechanical coupling both in the 2D plane and to their 3D environment both apically and basally. The last half of the manuscript is over-constrained by the authors early adoption of this theoretical framework. This adoption and the narrow interpretations of their experiments do not support the title nor the overbroad claims in the conclusions. This narrow perspective significantly reduced my enthusiasm for the entire manuscript.

Major problems:

The distributions of cell shapes across the embryo are well supported by junctional segmentation but the claims that cell shape change is simply a reflection of current strain fields is not supported - line 88. How do the authors know that these distributions were not created at earlier time points by asymmetric division or do not reflect intrinsic programs of cell shape driven by differential gene expression? One might alternatively assume that cell shapes were due to extant stresses and patterns of cell stiffness.

The authors make a statement that tissues near the serosal border are fluid because they rearrange but no other regions appear to be under such high stresses as those regions. If other regions, such as the dorsal side were put under similar stress would those cell not also rearrange?

As the authors mention, the use of a cell shape index to infer tissue mechanical properties is really only valid for a specific set of conditions. Have the authors confirmed those assumptions are met in the *Tribolium* serosa, both ventral and dorsal, across all the stages 0 to 5? Have the authors used other force inference methods? What do these alternative frameworks predict?

The authors interpret recoil velocities as indicators of tissue tension. This might be true when comparing single manipulations carried out on the same tissue at the same stage. However, the Hutson group (Ma et al, 2009, *Phys Biol*; Hutson et al, 2009, *Biophys J*) shows that recoil velocities are the product of both tension and viscoelastic material properties of the material around the laser ablation site. These material properties should not be confused with the mock "viscosity" exponential coefficient often used to fit the recoil data. For instance, recoil velocities in high tension - high stiffness tissues may appear the same as those in low tension / low stiffness tissues. The current conclusions are overstated without some measure of tissue mechanical properties between stages or from dorsal to ventral locations.

Based on laser ablation recoil velocities the authors claim tensions decreased with time (line 175), however, similar analysis in *Drosophila* ventral furrow closure revealed that the tissue were actually stiffening with time. Some support for this can be seen in the higher levels of actomyosin localizing in the apical junctions surrounding the serosal margin.

Upon viewing the figures (fig 2, 4, and 5) I am struck by the highly variable intensity of myosin and F-actin at the margin of the closing serosa. There is also a great deal of variability in the shape of cells along these boundaries. The claim that the cable is essential reminds me of early reports of dorsal closure in *Drosophila*. It would be useful if the authors could highlight alternative hypotheses and structural models that would be compatible or alternatives to their current working model.

Line 204 - The authors couple the absence of myosin contractility with rearrangement, failure of fluidization, and failure of serosal closure. This statement requires a leap of logic that are not tested explicitly in the experiments. Are these four phenomena really so tightly coupled? What are some of the other possible explanations?

During closure the authors report that myosin localization remains constant. How does cell density along the margin change and does the length of a cell margin on the boundary also change over time? If a cell, or several cells at the margin are ablated early does that alter the rate of closure or these other descriptors of and extracellular contractile bundle?

What does the margin of the serosa look like in cross-section? Is the myosin cable at the margin also an indicator of a boundary between inner- and outer-serosa cells? When the boundary appears to disappear is it simply 'rolling' inside? What happens to this junction at closure?

The authors speculate that the absence of cell division is mechanically responsive to high levels of stress. However, many cases of morphogenesis in the early embryo progress without significant cell proliferation. One alternative suggestion has been that cytokinesis would weaken the mechanical continuity of stresses that are required to drive large scale motions or that eruptions of cell division might weaken epithelial sheets as they are under the most stress.

The authors mention that the appearance of an actomyosin-rich cable is always indicative of tension but this is not so. Please take a look at the review by Sawyer et al, 2010 for a summary of other cases where enriched actomyosin cables can reflect tissue boundaries. The revisions over time by Kiehart and colleagues should serve as a warning to such broad claims. While they were initially convinced, and convinced the field, that the bundle of actomyosin at the margin of the amnioserosa in *Drosophila melanogaster* was a load bearing structure they later recanted and adopted a more integrative view of dorsal closure.

Minor issues:

It is hard to believe that the authors really believe that epiboly is evolutionarily conserved. The movements may be considered a common theme in the morphogenesis of multicellular organisms but I would strongly object to the suggestion that morphogenesis of the serosal epithelium is evolutionarily related to teleost EVL epiboly or amphibian epiboly, etc.

NCOMMS-19-38732-T "Regionalized tissue fluidization by an actomyosin cable is required for epithelial gap closure during insect gastrulation"

Response to the Reviewer's comments:

Reviewer #1 (Remarks to the Author):

Using confocal and state of the art light sheet live imaging, the authors provide here a concise and comprehensive description of cellular events during early morphogenetic movements of the extra-embryonic serosa in the flour beetle *Tribolium*. With the help of laser scissors and RNA interference, they functionally probe the role of tissue tension and an actin/myosin cable at the growing/closing edge of the serosa to demonstrate intriguing similarities with other morphogenetic processes including wound healing or epiboly in vertebrates.

This is a very solid paper of high relevance beyond insect development that in my view is very well suited for NatComms. I have only minor corrections/questions as detailed below.

We thank the Reviewer for their positive assessment of our work.

(1) One major finding of this work is that the actin/myosin cable is essential to close the serosa window, and that inhomogeneities in cable thickness function to determine which cells are evicted next from the growing edge. While I think this interpretation is very likely correct, there is a potential problem in their argument: they visualize the cable through injection of a capped RNA encoding Myosin-GFP. However, given cell volumes in the blastoderm stage (when cellularization occurs) seem not very uniform, it is conceivable that much of the cell-to-cell variability of Myosin-GFP at later stages could be due to variation in the number of RNA molecules taken up by these cells at an early stage, or be due to stochastic degradation of mRNA. I wonder if the authors have considered staining fixed embryos using crossreacting antibodies (or maybe in situ hybridization) to verify that also the endogenous myosin2 is unevenly expressed in the cells of the leading edge.

Reviewer rightly wonders whether the reported inhomogeneity in myosin II accumulation in the cable-forming cell edges is an artifact of labeling embryos with injected mRNA encoding the fluorescent myosin II reporter. This is actually a misunderstanding. As shown in the Supplementary Methods Table, the data on myosin heterogeneity in Figure 4H-I are from a transgenic *Tribolium* line ubiquitously expressing the Tc-sqh-eGFP reporter. We now clarified this in the main text (line 196),

the Figure 4 legend (line 688) and in the Methods section *Tribolium* rearing and stocks (line 293-303).

(2) Their finding that tissue tension is higher dorsally than ventrally does fit their concept of tissue fluidity but raises the question how the mechanical force is generated that places tension upon the dorsal serosa. About this I have two questions:

(2a) did they probe ventral tissue tension through cuts parallel to the cable as well as at 90° to the cable, and if so was there a difference?

Regarding the orientation of the tissue-level cuts, we have clarified in the revised manuscript that the tissue-level cuts were oriented perpendicular to the axis along which the cells of the serosa window margin are stretched (lines 133-134). Considering the positions in the embryo where these cuts are inflicted (Figure 3C,D), these cuts are oriented perpendicular to the midline both in the dorsal and ventral regions. Tissue cuts in other orientations are experiments that in principle could be performed. However, we decided not to perform these experiments because we doubt that cuts parallel to the axis along which the cells are stretched would alter the main conclusion of the paper about the regionalization of the serosa along the dorsal-ventral axis.

(2b) if the force pulling at the dorsal serosa is not (alone) exerted by the ventral serosa (including the actin/myosin cable), where else is this pulling force coming from? Do they think the dorsal serosa cells actively "crawl" along the vitelline membrane?

This question raised by the Reviewer is a very intriguing one. We speculate that the main source of the force that increases the tension in the expanding dorsal serosa is generated from the condensation of the adjoining embryonic primordium, maybe in concert with additional forces generated by active crawling of the serosa on the vitelline membrane as the Reviewer suggests. We have however not addressed these hypotheses experimentally and therefore decided not to include this speculation in the manuscript. Examining the active role of serosa cell crawling on the vitelline substrate in epibolic tissue expansion will be the subject of follow up studies as this is an attractive hypothesis supported by recent work in *Drosophila* imaginal discs (J. Bellaïche personal communication).

(3) Legend Fig. 3C, D; line 551: why are dorsal recoil speeds in C (stage 3) only half of those in D (same stage and location!)?

The reason for this discrepancy is that the ablations shown in Fig. 3C were performed with a 2-photon laser set-up, while the ablations shown in Fig. 3D were performed with a UV laser set-up. This information is now added in the Figure 3 legend (lines 625, 631, 637, 641). The two systems have differences in the localization of energy densities during the tissue ablations resulting in different recoil speeds.

List of minor issues:

Line 243: "Anna Giles" is misspelt (=> Gilles)

Corrected.

Legend Fig. 3E, F: maybe in the legend the authors should refer to the methods section where these indices are defined

We have added in the Figure 3 legend (lines 649, 655) that this information can be found in the Methods section Shape index analysis.

Legend Fig. 4A and elsewhere throughout the paper and supplemental info: "... injected with Tc-sqh-eGFP", Tc-sqh-eGFP should be italic to indicate that capped in vitro transcribed RNA, not protein was injected.

Corrected throughout the manuscript.

Legend Fig.4C: the time stamp color code of the cable (blue => purple => red => yellow => green) does not reflect the color bar (blue => green => red)

We modified the color of the lines and the color bar in Figure 4C to accurately reflect the temporal progression of the cable outline.

Legend Fig. 4 I: "... shown in (I)" should be (H)?

Corrected, thank you.

Legend Fig. S1B: " bright dots surrounding embryo" => it is in Fig. S1C where these white dots are visible

We transferred the following text from Figure legend S1B to S1C: "Bright dots surrounding the embryo correspond to the fluorescent beads used for image registration and fusion" (lines 743-745).

Legend Fig. S2A: remove "a"

Corrected, thank you.

Legend Fig. S2D: indicate, to which time point in C "zero" corresponds. Also, it seems that anisotropy has been normalized for each cell at time zero, this should be mentioned in the legend.

We have added the following text to Figure legend S2D (line 769-770): "Time-point 0 is different for each measured cell and corresponds to the stage when it gets evicted from the cable and its anisotropy is the highest."

Fig. S3: is the anisotropy at the poles (left and right edges of projections) real or an artifact of cartographic projections?

The cartographic projections are corrected for distortion introduced in cylindrical projections prior to measuring anisotropy and other quantities for each cell (described in lines 372-377 in Methods section Image processing). Minor remaining distortions at the edges corresponding to the poles of the embryo are largely due to the cells that are split when unfolding the continuous 3D shape to a 2D map.

Supplementary movie 4: some relevant cells have been color-highlighted to facilitate tracking. This in part obscures the original Life-Act-GFP signal, making it hard to actually see the original signal. I would like to see the original movie and suggest to also include in the supplemental material the unlabeled version of the movie.

We have generated a revised Supplementary movie 4 with the original movie on the left and the annotated color-coded tracked cells on the right.

Supplementary movie 11: it should be explained also in the legend, why there is a dorsal gap in

the projections

We have added the following text in the legend of Supplementary movie 11 (lines 906-908): “The edges of the cartographic projections (corresponding to the dorsal midline and poles of the embryos) that were excluded from our quantifications were also excluded from the 3D renderings.”

--

Reviewer #2 (Remarks to the Author):

In this manuscript Jain and co-workers have identified a novel mechanism driving epithelial cell rearrangement during epiboly in tribolium. Using a combination of quantitative imaging approaches and laser cut experiments they found that during epiboly, serosa cells are compartmentalized into two distinct territories displaying distinct mechanical properties. A dorsal region displaying more solid and elastic characteristic (as clearly demonstrated by the calculated recoil velocities upon laser cut), and a more ventral region that is more fluid and less elastic. They found that this behavior is determined by the assembly of an heterogenous contractile actomyosin at the leading edge of the serosa region that by contracting facilitates cell intercalation away from the margin. Although a similar mechanism has been recently described during wound healing in response to injury in other contexts, this study provides the first demonstration that such a mechanism underlies the regulation of a developmentally controlled morphogenetic process. In general, I find that the experiments are well conducted and properly controlled with detailed information and statistical analysis and I therefore support publication of this manuscript in Nature Comm.

We thank the Reviewer for their positive assessment of our work.

I have two suggestions which I think will help strengthen the conclusions of this study.

1) Given the focus of this manuscript on tissue fluidity, I think it would be important to perform “creep and recovery” measurements to directly measure differences in tissue fluidity either by pipette aspiration or by injection of magnetic beads (if at all possible in tribolium).

We agree with the Reviewer that these would be informative experiments that could strengthen and extend (but surely would not change) our main finding about the solid-like and fluid-like state of the dorsal and ventral serosa. Regarding micropipette aspiration, it has been used successfully to

measure viscoelasticity in cells, tissue explants and whole embryos of vertebrates, including *Xenopus*, chicken, zebrafish and mouse (e.g. reviewed in (Guevorkian and Maître, 2017)). However, similar microaspirations are not applicable in insect embryos, like *Drosophila* and *Tribolium*, where the outer protective vitelline membrane cannot be removed without perturbing normal development. In principle, injection of magnetic particles into insect embryos and measurement of their response to externally applied magnetic forces is a less invasive approach. It is a challenging methodology to apply *in vivo* and has been recently successfully used in *Drosophila* embryos but never in *Tribolium*. It requires a very specialized set-up that we currently do not have in our institutes and, thus, is not feasible for us to perform the experiments within a reasonable time frame.

2) The authors should explain better why they hypothesized that removal of *zerknüllt-1* would result in loss of the assembly of the actomyosin cable. They should also discuss that while the results of this experiment support their conclusion, the interpretation of this particular mutant phenotype might be more complex. *Zerknüllt-1* is a transcription factor expressed in all serosa cells and could impact on additional cellular behaviors that were not the focused of this analysis. It is tempting to suggest additional optogenetic experiments to more precisely dissect the role of actomyosin cable contraction in controlling tissue fluidization. However, I do appreciate that these experiments are not feasible at this point in *tribolium*.

The Reviewer raises a valid point. Perturbing the accumulation of actomyosin at a tissue boundary by optogenetic means *in vivo* is technically difficult in any system, including for example *Drosophila*, and attempting this in *Tribolium* goes beyond the scope of this study. Nevertheless, gene knock-down by RNAi is the most powerful technique in beetles that can be applied at all developmental stages and elicit a strong systemic response, often phenocopying null mutants. We have extended the text of the manuscript (lines 206-222) to better explain our rationale for using the power of *Tribolium* RNAi approach to disrupt a tissue boundary and thus functionally test our model for an actomyosin cable-mediated serosa tissue fluidization:

“Such a model predicts that in the absence of the actomyosin cable, the serosa window would fail to close ventrally. A previous study suggested that the juxtaposition of normally proportioned extraembryonic (serosa) and embryonic (amnion and germband) rudiments in the differentiated *Tribolium* blastoderm is required for proper emergence and constriction of the actomyosin cable at the extraembryonic/embryonic boundary (Benton et al., 2013). Furthermore, it has been demonstrated that the transcription factor-encoding *zerknüllt-1* gene (*Tc-zen1*) has an early function

in specifying serosal cell fate, and that RNAi knock-down of Tc-zen1 results in serosa-less embryos that are not covered by extraembryonic membranes ventrally (van der Zee et al., 2005). Based on this evidence, we hypothesized that Tc-zen1^{RNAi} embryos would be lacking the actomyosin cable. Although Tc-zen1 knock-down is expected to impact multiple cellular properties in the anterior blastoderm, where cells are transformed from serosal into embryonic (most likely amniotic) fate, Tc-zen1^{RNAi} embryos exhibit a very specific early morphogenetic defect without significantly compromising the morphology and viability of late embryos (van der Zee et al., 2005). Live imaging of transgenic embryos expressing LifeAct-eGFP obtained after parental knock-down of Tc-zen1 revealed indeed the absence of the actomyosin cable (Fig 5A,B, Supplementary movie 9). While such Tc-zen1^{RNAi} embryos started the contraction and folding of the embryonic primordium as wildtype embryos, the epibolic movement halted and a ventral serosa window failed to form and close (Fig 5A,B,E, Supplementary movie 10).”

--

Reviewer #3 (Remarks to the Author):

Jain et al. report on the cellular changes and anisotropies that occur during amnioserosa formation/function in Tribolium. They apply good quantitative skills and analysis to measuring cell shapes and tissue movements as the amnioserosa expands across the Tribolium embryo, and analogize these movements to epiboly processes in vertebrate organisms. They further interrogate tissue tensions through laser cuts. I found the results (especially the heterogeneity and analysis of the actomyosin cable) interesting, and I think it is solid work, but I am not sure the novelty or insight of the results rises to the level of Nature Communications. I think the current work lacks a certain mechanistic depth and is somewhat descriptive (some of the findings are unsurprising, at least to me), but this is not to say that it isn't highly meritorious. I also found the fluidization by intercalary movements not especially novel (again, nice for this system), but not profoundly insightful. For me, this would make for a very solid publication in Development – however, this is an admittedly subjective criteria and I am happy to hear if other reviewers have different opinions. The manuscript is well-written, and statistics are (largely) done appropriately. More detailed comments are below:

Thank you for acknowledging the quality of our manuscript. Reviewer #2 nicely spells out what would be also our first response to Reviewer's #3 concern about the novelty of our work that *“although a similar mechanism has been recently described during wound healing in response to*

injury in other contexts, this study provides the first demonstration that such a mechanism underlies the regulation of a developmentally controlled morphogenetic process". The original description of tissue dynamics during serosa epiboly in *Tribolium* and the potential involvement of an actomyosin cable in serosa window closure were indeed reported by one of us in the journal *Development* (Benton et al., 2013). This study goes several steps further and employs a combination of advanced *in toto* imaging and image analysis tools together with genetic and biophysical perturbations to provide a quantitative, mechanistic explanation of how serosa closure may be achieved: regionalization of this epithelium, hitherto considered uniform, into territories with distinct mechanical properties through actomyosin contractility at its leading edge increasing the rate of cell intercalation (i.e. increasing tissue fluidity). We are fundamentally interested in sharing these results with the scientific community and are less concerned about what particular publishing venue will lead to this goal. We however argue that the connection between actomyosin cable at tissue boundaries and tissue intercalation and fluidization in epiboly and the intriguing similarities with general wound healing mechanisms, will be appealing to a broader multidisciplinary audience of *Nature Communications*.

More detailed comments are below:

1) There was one problematic statement, which can probably be addressed textually, but also brings up one of the central weaknesses of the manuscript (lines 184-187), "Since the myosin intensity correlates with the cell leaving behavior, we conclude that differential line tension along the cable circumference drives the eviction of the cells from the cable and the resulting cell rearrangements lead to tissue fluidization and eventual closure of the epithelial gap" – this was just a correlation, but the authors have written a statement of causation, which can't be done based on the data to this point of the manuscript (could be rephrased as a hypothesis).

The Reviewer is right that the correlation between myosin intensity and the order of cell eviction was not tested experimentally. We have changed the word "conclude" with "hypothesize" in line 201 and have also accordingly modified the text in the last paragraph (line 246-247) as follows: "The order in which cells are evicted correlates with local myosin accumulation at each cable-forming edge."

I think the authors are setting up the zen RNAi experiments in the following paragraph, which are also problematic. The zen RNAi is interpreted as actin-cable-less embryos, but this is really a drastic cell fate transformation of the amnioserosa. For me, this is too multifaceted a change to

represent a clean testing of actomyosin cable function, and therefore a full testing of one of the main hypothesis of the manuscript is not performed. Could Y-27632 be introduced at various points to test Myosin function more acutely and specifically?

The Reviewer is correct. We have been setting up a linear narrative of the manuscript in order to present a concise and accessible message. We have now modified the transition to the *Tc-zen1^{RNAi}* experiments and the logic behind the *Tc-zen1^{RNAi}* experiment has been expanded in the main text and is detailed in our response to Reviewer #2 (lines 206-222). Considering the lack of tools for localized perturbations in *Tribolium*, this is in our opinion currently the “cleanest” experimental manipulation at our disposal to eliminate actomyosin contractility at the leading edge. In fact, we tried without success other approaches to perturb the cable, including Rho kinase inhibitor Y-27632 suggested by the Reviewer. Unlike in *Drosophila* where the drug is delivered in the yolk sac, underneath the cells to be perturbed, in the case of *Tribolium* there are three cell layers (the condensed embryonic primordium, the amnion and the serosa) in between the yolk and the cable. When we injected in the yolk, we saw arrested development and no clear phenotype. In addition, injection close to the cable in the space between the serosa and the amnion is not possible due to the flatness and tight apposition of the two epithelia (shown in the new **Supplementary Fig 7**). We also attempted knocking down the *Tribolium* Zasp52 gene, that in *Drosophila* leads to specific actomyosin cable perturbation (Ducuing and Vincent, 2016), however we did not observe any abnormal phenotypes in the beetle embryo.

2) The authors are mostly good at this, but statistical tests and n numbers should be listed in the figure legend for every figure panel where appropriate (some panels were missing this information). Please justify that data is normal and appropriate for parametric analysis where this is used.

We have now confirmed that all sample numbers are included in the Figure legends. We make it also clear in line 574 that “The normal distribution of the data was tested using the Shapiro-Wilk test” and indicate which non-parametric or parametric statistical test was used in the figure legends of all relevant panels.

3) A little more detail on the cell shape and Q shape method is needed – it was hard in reading the manuscript to fully understand what was going on in this section (without doing more reading of the Methods and background papers (which I did to a limited degree)). I know this is based off of

other labs' work, but I don't fully buy the applicability of these thresholds across epithelial systems and it seems that some validation in this system is needed.

As detailed below, we expanded the explanation of the theory in the main text of the manuscript to provide additional guidance to the general readership unfamiliar with this established theoretical framework. We note that this framework was applied recently by its original authors to the *Drosophila* germband epithelium (Wang et al., 2020) and, as we argue in response to Reviewer #4, it is highly applicable to the *Tribolium* system that represents in our opinion an even closer approximation of a flat epithelial sheet under tension compared to the *Drosophila* blastoderm.

Lines 111-117 “Movement of cells past each other during neighbor exchange has been linked to increased tissue fluidity¹⁸⁻²¹. A useful theoretical framework to assess the behavior of the serosal tissue is the shape index analysis that infers solid-like or fluid-like tissue states from cell shapes in epithelia²²⁻²⁴. Based on the vertex model, the leading theoretical framework for studying the mechanical behavior of epithelial tissues²⁵, the theory predicts a critical value of shape index $p = 3.81$ marking the transition from a solid-like ($p < 3.81$) to a fluid-like behavior ($p > 3.81$ but see also²⁶ and below).”

Lines 152-164 “While the recoil pattern after laser ablation supports the hypothesis of ventral tissue fluidization suggested by the shape index analysis, it has been recently shown that the relationship between shape index and tissue fluidity is non-linear when the tissue is under tension²⁶. Since we obtained from laser ablations evidence that the *Tribolium* serosa exhibits a spatially in-homogenous tension profile, we applied this extended theoretical framework. Moreover, we observed that the cells close to the window are strongly elongated in direction radial to the window (Fig 2E,F, Supplementary Fig 2) which could indicate local anisotropy in the tension profile. Therefore, we calculated a local cell alignment factor Q across the serosal tissue as a proxy measure of local tissue tension anisotropy (see Methods)²⁹. The theory predicts that for a given value of Q the shape index p needs to exceed an adjusted threshold value in order for the tissue to be fluid-like. For each local value of Q across the cartographic maps, we plotted the difference between the actual shape index value (p) of the cell and the local threshold signifying solid-to-fluid transition (Fig 3E,F, Supplementary Movie 11).”

Minor notes:

The writing really is nice, and the entire preparation of the manuscript is well done. The authors are to be complimented on a nice piece of work.

We thank the Reviewer for their appreciation of the quality of our work.

--

Reviewer #4:

Enabled by recent advances in light sheet imaging and analysis tools the authors provide an elegant 3D description of serosal epiboly in the short germ band model insect *Tribolium castaneum*. In toto imaging utilizes fluorescent-protein tagged nuclei and the authors use additional probes to reveal actomyosin localization at the margin of the serosa as it encloses the developing embryo. To explain these movements and the physical mechanics of closure the authors adopt recent foam-based theory. This theory posits that one can surmise the mechanical properties based exclusively on the perimeter to area ratio of cells. Cells with longer perimeters than an equivalent isodiametric cell are considered "fluid". This theory is well suited for 2D conditions that confluent cultured epithelial cells inhabit, when cell sheets are most like 2D foams, but quickly fail when cell sheets inhabit more dynamic environments, such as embryos where cells may be patterned with distinct identities, different mechanical properties, and different levels of mechanical coupling both in the 2D plane and to their 3D environment both apically and basally. The last half of the manuscript is over-constrained by the authors early adoption of this theoretical framework. This adoption and the narrow interpretations of their experiments do not support the title nor the overbroad claims in the conclusions. This narrow perspective significantly reduced my enthusiasm for the entire manuscript.

We agree with the Reviewer that many mechanical aspects influence the development of an embryo and that a variety of physical frameworks can be used to describe each of them. However, we would like to re-emphasize that the serosa epiboly we investigate in the current study, involves the spreading of a squamous (flat) epithelial sheet that does not exhibit cell divisions, and approximates closely an idealized 2D foam-like tissue, similar to monolayers in cell culture. Therefore, we specifically chose a theoretical framework that has been applied successfully not only to cultured monolayers, but also recently to the morphogenesis of insect epithelia (Wang et al., 2020). We regret that relying on this framework weakens the Reviewer's enthusiasm for our work, but we

maintain that this is the correct physical picture to use for understanding the underlying mechanics of serosa spreading.

Major problems:

The distributions of cell shapes across the embryo are well supported by junctional segmentation but the claims that cell shape change is simply a reflection of current strain fields is not supported - line 88. How do the authors know that these distributions were not created at earlier time points by asymmetric division or do not reflect intrinsic programs of cell shape driven by differential gene expression? One might alternatively assume that cell shapes were due to extant stresses and patterns of cell stiffness.

We would like to stress that there are no cell divisions in the extraembryonic serosal epithelium during the morphogenetic process we study here. Prior nuclear (not cellular) divisions occur within the syncytium and it is well established in multiple insect systems, including *Tribolium*, that cells are uniform when the tissue cellularizes. We start our investigation at this stage (stage 0) and measure that uniform distribution of cell shapes exists at this point (see Fig. 2G,I). Thus, we can rule out that asymmetric divisions have created the distributions of cell shapes we observe later when comparing dorsal and ventral parts of the serosal epithelium.

As for the possibility of differential gene expression, there has been extensive, near genome-wide studies of expression patterns at the blastoderm stage in *Tribolium* (Stappert et al., 2016). We have surveyed those patterns and failed to find a clear correlation with the regionalization of the serosa that we detected in our shape analysis. This does not mean that such underlying patterns of gene expression do not exist. Closing this multiscale loop completely, from patterns of gene expression through cell dynamics to tissue level mechanics would be indeed an amazing feat, but well beyond the scope of our manuscript.

To the third point, we tried to avoid any speculation as to the sources of the stresses that the serosa is demonstrably exposed to. We do not have data on the sources of stresses across the embryo nor on the patterns of tissue recoil beyond the serosa tissue.

The authors make a statement that tissues near the serosal border are fluid because they rearrange but no other regions appear to be under such high stresses as those regions. If other regions, such as the dorsal side were put under similar stress would those cell not also rearrange?

We do not know. We have no means to ectopically induce an actomyosin cable in the dorsal side of the embryo or physically manipulate the dorsal tissue. But yes, we would exactly expect what the Reviewer suggests, namely that if other regions of the tissue were put under similar stress like the serosa margin, these cells would exhibit rearrangements.

As extra support for this statement, we added in our response to Reviewers (but not in the paper) preliminary data on dorsalized *Tribolium* embryos induced by Tc-Toll1 RNAi. As previously described, Tc-Toll1^{RNAi} blastoderm embryos are subdivided into an expanded anterior serosa and a reduced posterior embryonic rudiment with a straight (instead of tilted in wildtype) border between them. These embryos gastrulate largely symmetric along the DV axis and close their serosa ectopically at the posterior pole (instead at the ventral side in wildtype) (see Suppl Figure S4 in (Nunes da Fonseca et al., 2008)). By imaging serosal cell dynamics in Tc-Toll1^{RNAi} embryos labeled with LifeAct-eGFP, we observed that the leading serosal cells were stretched along the anterior-posterior axis (instead of the dorsal-ventral axis in wildtype), were evicted from the cable and planarly intercalated anteriorly into the serosal epithelium (see Figure below).

Figure: Leading serosal cell dynamics in Tc-Toll1^{RNAi} embryos

Maximum intensity projections of a developing Tc-Toll1^{RNAi} embryo labeled with LifeAct-eGFP and imaged with a spinning disc confocal microscope. Arrows point to the cable surrounding the serosa window that closes ectopically near the posterior pole. Selected tracked cells at the leading edge of the serosa are outlined and colored to show that they shrink their cable-forming edges, they elongate orthogonally to the window along the anterior-posterior axis and intercalate anteriorly into the serosal tissue. Time-stamps are hh:mm. Scale bar is 100 μ m

As the authors mention, the use of a cell shape index to infer tissue mechanical properties is really only valid for a specific set of conditions. Have the authors confirmed those assumptions are met in the *Tribolium* serosa, both ventral and dorsal, across all the stages 0 to 5? Have the authors used other force inference methods? What do these alternative frameworks predict?

We mention that the threshold signifying fluid to solid transition depends on the amount of external stress the tissue is subjected to according to (Wang et al., 2020). The strain in the tissue is inferred from local cell axes alignment (Wang et al., 2020). We show that during closure (stages 2-4), when

the threshold is adjusted to the strain, the ventral cells closer to the window still exhibit a behavior that the theory classifies as fluid-like. We would like to point out that we are not insisting on a step-wise transition from a fluid to a solid region of the serosa. Rather we propose that there is a gradient of tissue fluidity from the ventral to the dorsal side at the same stage, induced by the inherent asymmetry of the ventral cable forming at the tissue boundary.

We corroborate this theory-based hypothesis with the laser cutting experiments. These experiments suggest that indeed there is a gradient of tissue properties (whatever they may be, based on recoil velocities as we discuss below) from ventral to dorsal side of the embryo. We would like to point out that published work on wound healing infers local tissue fluidization simply from the detection of cell intercalation events (Tetley et al., 2019). In that sense, our work goes a step further in showing that there is a difference in fluidity across the tissue, fits a theoretical prediction for monolayer epithelia and demonstrates different tension profiles.

The authors interpret recoil velocities as indicators of tissue tension. This might be true when comparing single manipulations carried out on the same tissue at the same stage. However, the Hutson group (Ma et al, 2009, Phys Biol; Hutson et al, 2009, Biophys J) shows that recoil velocities are the product of both tension and viscoelastic material properties of the material around the laser ablation site. These material properties should not to be confused with the mock "viscosity" exponential coefficient often used to fit the recoil data. For instance, recoil velocities in high tension - high stiffness tissues may appear the same as those in low tension / low stiffness tissues. The current conclusions are overstated without some measure of tissue mechanical properties between stages or from dorsal to ventral locations.

We agree with the Reviewer that **"interpretations of recoil velocities as indicators of tissue tension might be true when comparing single manipulations carried out on the same tissue at the same stage"**. Indeed, we are using recoil velocities as an indicator of tension when comparing dorsal and ventral regions on the same tissue (serosa) and at the same stage (Stage 3, Fig 3D). The Reviewer rightly points out that the measured recoil velocities after laser cuts are influenced by both tension and viscoelastic material properties of the ablated tissue. This raises the possibility that there are regional differences in viscoelastic properties of the serosa tissue at a given stage and/or that they change over time. We accept the point raised by the Reviewer that it is not trivial to disentangle the contribution of tension and material properties based on laser cutting experiments alone. We argue however, that from all available methods to probe cell and tissue mechanics (reviewed in (Campàs, 2016)), the laser ablation approach employed in our study is currently the only available option in

Tribolium embryos. Therefore, we use it with the acknowledged caveat that serosa tension can be only inferred under the explicit assumption (line 144-145) that tissue material properties do not vary significantly as is commonly done in almost all developmental studies employing laser cutting.

Considering these limitations, in an attempt to address the Reviewer's concern with the tools and data at our disposal, we have now extended the analysis of the laser cuts to estimate the tissue properties from the dynamics of the tissue recoil, based on the methodology suggested by the Reviewer (many thanks). The results show that the ventral serosa recoils as a more fluid-like material compared to the dorsal serosa (as shown in new Supplementary Figure 4). We present an extensive rewrite of this crucial portion of the manuscript that we hope presents a more balanced view of the data:

Lines 138-143: "Because the tissue recoil velocity depends on both the tension in the tissue and the material properties of the tissue, we further analyzed the time-dependent decay of the tissue recoil²⁸. Such an analysis can discern between fluid-like or solid-like recoil patterns of the severed tissue. Our data suggested that the ventral tissue exhibits more fluid-like behavior than the dorsal tissue (Supplementary Fig 4 and Methods)."

Lines 352-359: "In an attempt to estimate tissue material properties from recoil dynamics after laser ablation, power-law analysis was performed as previously described²⁸. Briefly, a kymograph perpendicular to the laser cuts was obtained with the Multi-Kymograph plugin in Fiji [https://imagej.net/Multi_Kymograph]. The recoil velocity was quantified during the first 10 seconds after the ablation and fitted by a power-law $D(t-t_0)^\alpha$, where t_0 is the time of the ablation and D and α are fitting parameters. D correlates with the recoil velocities of the edges and α can acquire values from 0 to 1. A tissue with a value of α closer to 0 shows properties of an elastic solid, while a value closer to 1 indicates viscous fluid properties."

We have added a new author, Marina Cuenca, who performed this power-law analysis.

Based on laser ablation recoil velocities the authors claim tensions decreased with time (line 175), however, similar analysis in Drosophila ventral furrow closure revealed that the tissue were actually stiffening with time. Some support for this can be seen in the higher levels of actomyosin localizing in the apical junctions surrounding the serosal margin.

This must be a misunderstanding, since we report that the tension increases over time (lines 134-136 and Fig 3C). Extending the arguments mentioned above, we are reluctant to compare

mechanical properties across organisms. It may very well be that the mechanical properties of ventral furrow in *Drosophila* and serosa in the beetle are different.

Upon viewing the figures (fig 2, 4, and 5) I am struck by the highly variable intensity of myosin and F-actin at the margin of the closing serosa. There is also a great deal of variability in the shape of cells along these boundaries. The claim that the cable is essential reminds me of early reports of dorsal closure in *Drosophila*. It would be useful if the authors could highlight alternative hypotheses and structural models that would be compatible or alternatives to their current working model.

We are very happy that the Reviewer noticed the striking heterogeneity of myosin and F-actin intensity at the serosa margin. This observation is indeed central to our argument regarding the mechanism of tissue fluidization. Similarly, the shape variability at the margin is the intuitive representation of the peculiar shape transitions that we described and quantified extensively and that underlie the proposed regionalization of the serosa.

An alternative explanation, and one we do not favor, is that the cable constriction is the driving force for the window closure. We view the cable rather as linked chain of independently contracting cells (suggested by the triple cut experiments, lines 192-195 and Fig 4G) that due to its heterogeneity unjams the tissue at the margin by allowing cells to leave sequentially. This propagates into the tissue as an apparent difference in intercalation behavior and tension that we, based on theory, label as fluidization.

There are many sources of force generation besides the cable that affect positively or negatively the window closure: (i) the contraction of the with-serosa-continuous embryonic rudiment, (ii) the active crawling of serosa cells on the vitelline ((Münster et al., 2019) and unpublished observations) and (iii) the coupling to underlying yolk that exhibits strong autonomous actomyosin mediated contractility (unpublished observations). These interesting speculations will be subject to our future comprehensive study of yolk/blastoderm/envelope ensemble in *Drosophila*, *Tribolium* and other insects.

Line 204 - The authors couple the absence of myosin contractility with rearrangement, failure of fluidization, and failure of serosal closure. This statement requires a leap of logic that are not tested explicitly in the experiments. Are these four phenomena really so tightly coupled? What are some of the other possible explanations?

We acknowledge that the connection between myosin heterogeneity and cell eviction is correlative and that removal of the cable by changing serosa cell fate most likely has pleiotropic consequences (for instance, the remaining flat, dorsal cells may have reached their limit of stretchability and this could contribute to the halt of the window closure). We therefore softened the strong conclusion of the paragraph, as detailed in our response to the same point raised by Reviewer #3.

During closure the authors report that myosin localization remains constant. How does cell density along the margin change and does the length of a cell margin on the boundary also change over time?

These questions are addressed throughout our manuscript:

First, we report that (lines 94-97) “Our cell tracking experiments showed that the initial number of approximately 75 leading cells progressively decreased to only 5-6 cells during final serosa closure (Fig 2B,C) and that these cells originated from all around the periphery of the window (Fig 2D, Supplementary movie 3).”

Second, we describe that (lines 100-102) “The leaving cells shrunk their leading edge facing the serosa window and elongated radially in the direction approximately orthogonal to the window (Supplementary Figure 2A-C).” and that (lines 604-606) “Selected tracked cells at the leading edge of the serosa window are outlined and colored to show that they shrink their serosa-window-facing edges and planarly intercalate into the serosal epithelium (Figure legend 2E)”

Third, we measure that (lines 187-189) “As the cable shrunk, the total myosin intensity normalized by cable length stayed the same or increased over time (Supplementary Fig 6E).”

Finally, we observed that (lines 198-200) “Cells with more myosin contracted their cable-forming edges and were evicted from the leading edge of the serosa earlier than cells with lower levels of myosin (Fig 4H,I, Supplementary movie 8).”

If a cell, or several cells at the margin are ablated early does that alter the rate of closure or these other descriptors of and extracellular contractile bundle?

We do not understand what the Reviewer means when referring to an “extracellular contractile bundle”. When cells are ablated early, a wound is created and the progress of the window slows down.

What does the margin of the serosa look like in cross-section?

A detailed structural analysis of the *Tribolium* serosa has been carried out by Handel et al. ((Handel et al., 2000) using scanning electron microscopy. We are citing this paper extensively in the beginning of our manuscript (lines 67, 92 and 94) and are providing more detailed images of the serosa margin (including annotated schematic illustrations and cross-sections) in the new Supplementary Fig 7.

Is the myosin cable at the margin also an indicator of a boundary between inner- and outer-serosa cells? When the boundary appears to disappear is it simply 'rolling' inside? What happens to this junction at closure?

We are answering some of these questions in the new Supplementary Fig 7. As stated in the legend (lines 827-841), “the actomyosin enrichment is detected at the leading edge of the serosa” and “the squamous serosa bents inwards over the serosa window”. Upon window closure, the serosa and amnion separate into discrete membranes. This process has not been the focus of this work but has been described extensively elsewhere (Benton et al., 2013; Handel et al., 2000; Hilbrant et al., 2016; Koelzer et al., 2014).

The authors speculate that the absence of cell division is mechanically responsive to high levels of stress. However, many cases of morphogenesis in the early embryo progress without significant cell proliferation. One alternative suggestion has been that cytokinesis would weaken the mechanical continuity of stresses that are required to drive large scale motions or that eruptions of cell division might weaken epithelial sheets as they are under the most stress.

The lack of cell division in the serosa is simply a fact. We do not claim that this increases stress. On the contrary, we agree with the Reviewer that in many developmental scenarios bursts in cytokinesis weaken mechanical stresses. As this mechanism of stress relief is not available in *Tribolium* serosa, we sought for an alternative explanation.

The authors mention that the appearance of an actomyosin-rich cable is always indicative of tension but this is not so. Please take a look at the review by Sawyer et al, 2010 for a summary of other cases where enriched actomyosin cables can reflect tissue boundaries. The revisions over

time by Kiehart and colleagues should serve as a warning to such broad claims. While they were initially convinced, and convinced the field, that the bundle of actomyosin at the margin of the amnioserosa in *Drosophila melanogaster* was a load bearing structure they later recanted and adopted a more integrative view of dorsal closure.

We measured the tension in the cable and the data show that a) it increases over time (Fig 4D, E, F) and b) it is independent from cell to cell (Fig 4G). As stated before, we do not view the margin as a load bearing structure. In fact, our view appears to be aligned with that of the Reviewer. We agree that for a complete understanding of the mechanical forces that lead to serosa window closure, we need to adopt an integrative approach that takes into account tissue-intrinsic force generation across the blastoderm and, in addition, the interactions between the cells of the tissue and the vitelline envelope and the yolk. As a first step toward this goal, we addressed here the role of actomyosin cable as a stress-releasing mechanism in a non-dividing epithelium closing on an ellipsoid. The proposed mechanism of cable induced tissue fluidization during *Tribolium* serosa closure has broad implications for epithelial gap closure in development and during tissue repair.

Minor issues:

It is hard to believe that the authors really believe that epiboly is evolutionarily conserved. The movements may be considered a common theme in the morphogenesis of multicellular organisms but I would strongly object to the suggestion that morphogenesis of the serosal epithelium is evolutionarily related to teleost EVL epiboly or amphibian epiboly, etc.

If the Reviewer is referring to the very first sentence of the manuscript main text (line 46), we are more than happy to change the wording to “Epiboly is one of the hallmark morphogenetic movements”. Indeed, epibolic movements during insect, fish and amphibian gastrulation all involve tissue spreading and thinning, and are realized by partly overlapping but also very different mechanisms. In order to be able to say something definitive about its evolution, we need to study many different realizations of this morphogenetic module in many species, understand its regulation and function, map the data onto the animal phylogeny and use quantitative evolutionary inference to understand its origin and diversity.

References:

- Benton, M.A., Akam, M., and Pavlopoulos, A. (2013). Cell and tissue dynamics during *Tribolium* embryogenesis revealed by versatile fluorescence labeling approaches. *Development* *140*, 3210–3220.
- Campàs, O. (2016). A toolbox to explore the mechanics of living embryonic tissues. *Semin. Cell Dev. Biol.* *55*, 119–130.
- Ducuing, A., and Vincent, S. (2016). The actin cable is dispensable in directing dorsal closure dynamics but neutralizes mechanical stress to prevent scarring in the *Drosophila* embryo. *Nat. Cell Biol.* *18*, 1149–1160.
- Guevorkian, K., and Maître, J.-L. (2017). Micropipette aspiration. In *Methods in Cell Biology*, (Elsevier), pp. 187–201.
- Handel, K., Grünfelder, C.G., Roth, S., and Sander, K. (2000). *Tribolium* embryogenesis: a SEM study of cell shapes and movements from blastoderm to serosal closure. *Dev. Genes Evol.* *210*, 167–179.
- Hilbrant, M., Horn, T., Koelzer, S., and Panfilio, K.A. (2016). The beetle amnion and serosa functionally interact as apposed epithelia. *Elife* *5*, 14217.
- Koelzer, S., Kölsch, Y., and Panfilio, K.A. (2014). Visualizing Late Insect Embryogenesis: Extraembryonic and Mesodermal Enhancer Trap Expression in the Beetle *Tribolium castaneum*. *PLOS ONE* *9*, e103967.
- Münster, S., Jain, A., Mietke, A., Pavlopoulos, A., Grill, S.W., and Tomancak, P. (2019). Attachment of the blastoderm to the vitelline envelope affects gastrulation of insects. *Nature* *568*, 395–399.
- Nunes da Fonseca, R., von Levetzow, C., Kalscheuer, P., Basal, A., van der Zee, M., and Roth, S. (2008). Self-Regulatory Circuits in Dorsoventral Axis Formation of the Short-Germ Beetle *Tribolium castaneum*. *Dev. Cell* *14*, 605–615.
- Stappert, D., Frey, N., von Levetzow, C., and Roth, S. (2016). Genome-wide identification of *Tribolium* dorsoventral patterning genes. *Development* *143*, 2443–2454.
- Tetley, R.J., Staddon, M.F., Heller, D., Hoppe, A., Banerjee, S., and Mao, Y. (2019). Tissue fluidity promotes epithelial wound healing. *Nat. Phys.*
- Wang, X., Merkel, M., Sutter, L.B., Erdemci-Tandogan, G., Manning, M.L., and Kasza, K.E. (2020). Anisotropy links cell shapes to tissue flow during convergent extension. *Proc. Natl. Acad. Sci.* *51*, 201916418.
- van der Zee, M., Berns, N., and Roth, S. (2005). Distinct Functions of the *Tribolium* *zerknullt* Genes in Serosa Specification and Dorsal Closure. *Curr. Biol.* *15*, 624–636.

Reviewers' Comments:

Reviewer #1:

Remarks to the Author:

I am generally happy with the answers to the points raised in my review, and remain convinced that this paper merits publication in NatComms.

Concerning the issues raised by the other three reviewers I very much appreciate the analytic depth they invested in questioning the paper's assumptions, proposing additional experiments, and forcing the authors to even more critically distinguish between concepts proven or just suggested by their work. Through this the text has been significantly sharpened in this revision, and several experiments have been added.

Many proposed experiments could not be included in the revised paper, and I would like to support the authors here: *Tribolium* still is an emerging model system, and importing techniques from *Drosophila* or other models is not quite as straightforward as some might think. Years of additional effort will be required to establish techniques like optogenetics, direct tissue fluidity measurements, or new methods to specifically interfere with forces potentially generated by the different embryonic structures present at this stage of development.

Using available methods, the authors have made with this paper a big step towards understanding the soft matter mechanics of early embryogenesis in insects, with sufficient relevance for other systems to be of interest to a broad readership.

Reviewer #2:

Remarks to the Author:

The authors have addressed my concerns and I recommend publication. The role of actomyosin contractility and tissue fluidisation in morphogenesis is timely and therefore this study will be of interest to the community and will stimulate more work in other systems as well. Obviously not all the experiments that are possible in model organisms like *Drosophila* are possible in *Tribolium* at the moment. Overall the authors have convincingly revised the manuscript to incorporate reviewers' feedback and the data presented support the conclusions.

Stefano De Renzis

Reviewer #3:

Remarks to the Author:

Jain et al. report on the cellular changes and anisotropies that occur during amnioserosa formation/function in *Tribolium*. They apply good quantitative skills and analysis to measuring cell shapes and tissue movements as the amnioserosa expands across the *Tribolium* embryo, and analogize these movements to epiboly processes in vertebrate organisms. The authors also seriously reply to the reviewer concerns – I thank the authors for the care they illustrate in the resubmission letter. Unfortunately, I continue to think that it is a solid piece of work, but I don't find that the novelty or insight of the results rises to the level of Nature Communications. For me, most of the findings are unsurprising (again, this is not to say that they are not meritorious), and the level of analysis does not represent a substantial step forward (the analysis is well-done, but does not contribute towards a potential high-impact of the work). The most interesting portions are the observations on Myosin II heterogeneity, although this is complicated by the difficulties in addressing Myosin function in their system. These difficulties are understandable, but still mean a clean disruption of Myosin II, and thus a true testing of Myosin cable function, which is the crux of the manuscript, is

not possible. The authors have addressed some of the minor issues raised by this reviewer (and others), and have nicely edited some of the language of causation vs correlation. My apologies, but as I stated before, I believe this would be solid publication in *Development*. The edited writing in the manuscript represents an improvement, and the added statistical analyses and n-numbers are done appropriately.

Reviewer #4:

Remarks to the Author:

The authors have clarified several aspects of their study. The description of gastrulation, the cartographic mapping, and the cell shape analysis is very high quality and will make a strong contribution to the field of epithelial morphogenesis. However, I am still troubled by some of the responses to reviewers' questions and I remain unsatisfied with the mechanistic analysis, especially use of the term viscosity and the interpretation of the laser ablation studies.

In their response to reviewer #1, point #2a, the authors state that they did not make cuts parallel to the assumed line of tension. The whole point of the experiment was to test this assumption. Cell alignment can be due to a range of stress patterns, not all of them involve tension in only one direction. These cells may experience tension or compression in the orthogonal direction. The ability of cells to support compression would suggest that they are not fluidizing during closure.

In point #2b, there is yet another alternative to a traction-mediated crawling process, in that cells may simply be softening (see West et al, *Curr Bio* 2017).

The authors response to point #3 that the power of lasers used for ablation impacts recoil velocity is troubling. The instantaneous and complete cut of a tensional load-bearing structure is a key assumption when using this tool. The power of the laser, whether 2-photon high intensity infrared or 1-photon UV, should produce the same result.

I remain unconvinced by the claims of the second half of the manuscript and thus the title and conclusions. The authors mix time scales in a rather disingenuous manner. The time scale recoil after laser ablation is considerably shorter than the time scale of cell rearrangement. While the term 'viscosity' can be applied to these two processes, measurements of viscosity of recoil need not reflect the rate of cell rearrangement during closure. While laser ablation may be the only practical tool for exploring mechanics in blastoderm of *Tribolium* it is a blunt one that can only unambiguously report recoil velocities. The authors new analysis, e.g. fitting a power-law response to laser ablation and recoil, is very interesting but appears deeply flawed in their interpretation. The exponential time coefficient of the compliance, $\alpha > 0.75$, is significantly more fluid-like than any other tissue measured ($\alpha \sim 0.3$). However, this value for α is common to F-actin networks. This suggests then that the mechanical properties measured by laser ablation do not reflect tissue tension but rather reflects tension within the intracellular f-actin network (see de Sousa, J.S., Freire, R.S., Sousa, F.D. et al. (2020)).

NCOMMS-19-38732B "Regionalized tissue fluidization is required for epithelial gap closure during insect gastrulation"

Response to the Reviewer's comments:

Reviewer #1 (Remarks to the Author):

I am generally happy with the answers to the points raised in my review, and remain convinced that this paper merits publication in NatComms.

Concerning the issues raised by the other three reviewers I very much appreciate the analytic depth they invested in questioning the paper's assumptions, proposing additional experiments, and forcing the authors to even more critically distinguish between concepts proven or just suggested by their work. Through this the text has been significantly sharpened in this revision, and several experiments have been added.

Many proposed experiments could not be included in the revised paper, and I would like to support the authors here: *Tribolium* still is an emerging model system, and importing techniques from *Drosophila* or other models is not quite as straightforward as some might think. Years of additional effort will be required to establish techniques like optogenetics, direct tissue fluidity measurements, or new methods to specifically interfere with forces potentially generated by the different embryonic structures present at this stage of development.

Using available methods, the authors have made with this paper a big step towards understanding the soft matter mechanics of early embryogenesis in insects, with sufficient relevance for other systems to be of interest to a broad readership.

We thank Reviewer #1 for the positive assessment of our work and for acknowledging that we reached the boundaries of what is currently experimentally feasible in an emerging model organism like *Tribolium*.

--

Reviewer #2 (Remarks to the Author):

The authors have addressed my concerns and I recommend publication. The role of actomyosin contractility and tissue fluidisation in morphogenesis is timely and therefore this study will be of interest to the community and will stimulate more work in other systems as well. Obviously not all the experiments that are possible in model organisms like *Drosophila* are possible in *Tribolium* at the moment. Overall the authors have convincingly revised the manuscript to incorporate reviewers' feedback and the data presented support the conclusions.

We also thank Reviewer #2 for the positive assessment of our work and for appreciating the broad relevance of tissue fluidisation by actomyosin contractility in developmental morphogenetic programs.

--

Reviewer #3 (Remarks to the Author):

Jain et al. report on the cellular changes and anisotropies that occur during

amnioserosa formation/function in *Tribolium*. They apply good quantitative skills and analysis to measuring cell shapes and tissue movements as the amnioserosa expands across the *Tribolium* embryo, and analogize these movements to epiboly processes in vertebrate organisms. The authors also seriously reply to the reviewer concerns – I thank the authors for the care they illustrate in the resubmission letter. Unfortunately, I continue to think that it is a solid piece of work, but I don't find that the novelty or insight of the results rises to the level of Nature Communications. For me, most of the findings are unsurprising (again, this is not to say that they are not meritorious), and the level of analysis does not represent a substantial step forward (the analysis is well-done, but does not contribute towards a potential high-impact of the work). The most interesting portions are the observations on Myosin II heterogeneity, although this is complicated by the difficulties in addressing Myosin function in their system. These difficulties are understandable, but still mean a clean disruption of Myosin II, and thus a true testing of Myosin cable function, which is the crux of the manuscript, is not possible. The authors have addressed some of the minor issues raised by this reviewer (and others), and have nicely edited some of the language of causation vs correlation. My apologies, but as I stated before, I believe this would be solid publication in Development. The edited writing in the manuscript represents an improvement, and the added statistical analyses and n-numbers are done appropriately.

We would like to thank Reviewer #3 for appreciating the quality and rigor of this work and our sincere efforts to address all Reviewers' comments in the best possible way. The Reviewer has a valid point that the functional testing of myosin II accumulation by "cleaner" localized perturbations as in more advanced experimental models like *Drosophila* is still pending in *Tribolium*. In appreciation of this limitation, we have changed the wording about actomyosin generated tension in *Tribolium* wherever necessary in the text, including the title and abstract, to distinguish between correlative and causal associations. We removed "by an actomyosin cable" from the title and the explicit mention of tension from the abstract and have toned down the sentence about the actomyosin cable in the abstract as follows: "Our results suggest that a heterogeneous actomyosin cable contributes to the fluidization of the leading edge by driving sequential eviction and intercalation of individual cells away from the serosa margin". Finally, regarding the appropriate journal for publishing this work, we hold on our argument that the connection between actomyosin contractility, cell intercalation and tissue fluidization during epiboly, as well as the intriguing similarities with general wound healing mechanisms, will be appealing to the broader multidisciplinary audience of Nature Communications.

--

Reviewer #4 (Remarks to the Author):

The authors have clarified several aspects of their study. The description of gastrulation, the cartographic mapping, and the cell shape analysis is very high quality and will make a strong contribution to the field of epithelial morphogenesis. However, I am still troubled by some of the responses to reviewers' questions and I remain unsatisfied with the mechanistic analysis, especially use of the term viscosity and the interpretation of the laser ablation studies.

We thank Reviewer #4 for acknowledging the quality of our work and the impact it will have on the developmental morphogenesis field and address below her/his remaining concerns.

In their response to reviewer #1, point #2a, the authors state that they did not make cuts parallel to the assumed line of tension. The whole point of the experiment was to test this assumption. Cell alignment can be due to a range of stress patterns, not all of them involve tension in only one direction. These cells may experience tension or compression in the orthogonal direction. The ability of cells to support compression would suggest that they are not fluidizing during closure.

In point #2b, there is yet another alternative to a traction-mediated crawling process, in that cells may simply be softening (see West et al, Curr Bio 2017).

We have addressed these points by adding a new paragraph in the Discussion clarifying that in this study we focused on serosa regionalization along the dorsal-ventral axis and discussing other possible sources of forces driving serosa epiboly:

“While here we focused on the gradient of serosa properties along the dorsal ventral axis, the pulling of the epithelial sheet over an ovoidal shape is an inherently three-dimensional process and thus the tissue likely experiences stresses in many directions. More systematic probing of the mechanical properties of gastrulating *Tribolium* embryos will be required to understand the source of the forces acting on the serosa. We expect that the condensation of the adjoining embryonic primordium^{13,14}, together with additional forces generated by the attachment of the blastoderm to the vitelline envelope³⁰, the yolk¹³, active crawling of the serosa on the vitelline envelope¹² and regulated changes in the shape and stiffness of the serosal cells^{17,34,35} may also contribute to the serosal epiboly.”

The authors response to point #3 that the power of lasers used for ablation impacts recoil velocity is troubling. The instantaneous and complete cut of a tensional load-bearing structure is a key assumption when using this tool. The power of the laser, whether 2-photon high intensity infrared or 1-photon UV, should produce the same result.

We disagree with this point. It has been reported that UV and 2-photon laser ablations can produce different recoil velocities in the very same tissue and developmental stage (for example compare Rauzi et al. (2008) [1] with Fernandez-Gonzalez et al. (2009) [2]). We quote from Fernandez-Gonzalez et al. (pp. 739-740):

“These findings contrast with a previous report using an infrared laser (Rauzi et al., 2008). Low-power infrared ablation induces a lesion that is smaller (<0.2 mm) than the lesion induced by UV ablation in this study (1 mm), and generates retraction velocities that are several times slower (0.11 mm/s versus 0.5 mm/s reported here), suggesting that the infrared ablation experiments only partially disrupt the actomyosin network.”

To clarify this point, we have added the following statement in the Methods section Laser ablations:

“The observed differences in absolute recoil velocities between the UV and 2-photon laser ablation set-ups used in this study have been observed before and are thought to reflect difference in the extent of disruption of the actomyosin network^{36,49}. Keeping this in mind, we only compared recoil velocities generated with the same laser ablation set-up, as shown in Fig 3C,D.”

I remain unconvinced by the claims of the second half of the manuscript and thus the title and conclusions. The authors mix time scales in a rather disingenuous manner. The time scale recoil after laser ablation is considerably shorter than the time scale of cell

rearrangement. While the term 'viscosity' can be applied to these two processes, measurements of viscosity of recoil need not reflect the rate of cell rearrangement during closure. While laser ablation may be the only practical tool for exploring mechanics in blastoderm of *Tribolium* it is a blunt one that can only unambiguously report recoil velocities. The authors new analysis, e.g. fitting a power-law response to laser ablation and recoil, is very interesting but appears deeply flawed in their interpretation. The exponential time coefficient of the compliance, $\alpha > 0.75$, is significantly more fluid-like than any other tissue measured ($\alpha \sim 0.3$). However, this value for α is common to F-actin networks. This suggests then that the mechanical properties measured by laser ablation do not reflect tissue tension but rather reflects tension within the intracellular f-actin network (see de Sousa, J.S., Freire, R.S., Sousa, F.D. et al. (2020)).

Obviously, we agree with the Reviewer that recoil movements in response to laser cuts are much faster than cell rearrangements (seconds vs. minutes), and that the respective viscosities associated with these processes must not necessarily be connected. In fact, we do not use the term 'viscosity' anywhere in the manuscript text. We argue on the basis of the degree to which the tissue is solid-like or fluid-like (reflected by the parameter α in the power-law analysis). This is different from just an increase or decrease in viscosity of a fluid. We have inserted the following sentence about alpha value to the Methods section:

“... α gives an indication of the mechanical properties of the tissue and can acquire values from 0 to 1. A tissue with a value of α closer to 0 shows properties of an elastic solid, while a value closer to 1 indicates viscous fluid properties.”

Alpha can be considered an absolute reference to draw conclusions about mechanical properties across different experimental systems and stages. For example, α values measured in the *Drosophila* amnioserosa during dorsal closure are stage-dependent; the mean value of α decreases from ~ 0.4 to ~ 0.3 as the amnioserosa becomes more solid-like during development (Ma X. et al. 2009 [3]). In this study, while mean α is 0.3-0.4, α values range from 0.1 to 0.7. In the work mentioned by Reviewer #4 (de Sousa J.S. et al 2020 [4]), the authors use atomic force microscopy (not laser ablation) to probe the mechanical properties of the actin network in human cells (normal and cancerous) and find α values comparable to our serosa measurements. Extensive laser cuts - as we have performed - sever several lateral plasma membranes, removing their elastic contributions to the tissue, which leaves only the underlying actin network as the major load-bearing structure of the tissue. Thus, it is reasonable that the α values determined after extensive laser cuts can approach the limits measured for intracellular F-actin networks (de Sousa J.S. et al. 2020).

References

- 1) Rauzi M., Verant P., Lecuit T. and Lenne P. (2008) Nature and anisotropy of cortical forces orienting *Drosophila* tissue morphogenesis. *Nat Cell Biol.* 2008 Dec;10(12):1401-10.
- 2) Fernandez-Gonzalez R., Simoes S., Röper J., Eaton S. and Zallen J.A. (2009) Myosin II dynamics are regulated by tension in intercalating cells. *Dev Cell.* 2009 Nov;17(5):736-43.
- 3) Ma X., Lynch H.E., Scully P.C. and Hutson M.S. (2009) Probing embryonic tissue mechanics with laser hole drilling. *Phys Biol.* 6(3):036004.
- 4) de Sousa, J.S., Freire, R.S., Sousa, F.D. et al. (2020) Double power-law viscoelastic relaxation of living cells encodes motility trends. *Sci Rep* 10, 4749